# Mitigating Object Hallucination in Large Vision-Language Models via Image-Grounded Guidance

## Abstract

The advancement of Large Vision-Language Models (LVLMs) has increasingly highlighted the critical issue of their tendency to hallucinate non-existing objects in the images. To address this issue, previous works focused on using specially curated datasets or powerful LLMs (e.g., GPT-3.5) to rectify the outputs of LVLMs. However, these approaches require either expensive training/fine-tuning or API access to advanced LLMs for post-generation correction. In response to these limitations, we propose **M**itigating hallucin**A**tion via image-g**R**ounded gu**I**da**N**c**E** (`MARINE`), a framework that is both *training-free* and *API-free*. `MARINE` effectively and efficiently reduces object hallucinations during inference by introducing image-grounded guidance to LVLMs. This is achieved by leveraging open-source vision models to extract object-level information, thereby enhancing the precision of LVLM-generated content. Our framework's flexibility further allows for the integration of multiple vision models, enabling more reliable and robust object-level guidance. Through comprehensive evaluations across 5 popular LVLMs with diverse evaluation metrics and benchmarks, we demonstrate the effectiveness of `MARINE`, which even outperforms existing fine-tuning-based methods. Remarkably, it reduces hallucinations consistently in GPT-4V-assisted evaluation while maintaining the detailedness of LVLMs' generations.

## 1 Introduction

The advent of Large Language Models (LLMs) has motivated advancements in extending their remarkable capabilities to multimodal data. Grounded in the development of pre-trained vision-language models (Radford et al., 2021; Jia et al., 2021; Alayrac et al., 2022) that align visual and textual embedding spaces, Large Vision Language Models (LVLMs) have gained substantial attention in both architectural development (Liu et al., 2023d; Zhu et al., 2023; Ye et al., 2023; Dai et al., 2023a; Gao et al., 2023), alignment (Yu et al., 2024; Zhou et al., 2024; Deng et al., 2024) and benchmarking datasets (Xu et al., 2023; Lu et al., 2024; Zhang et al., 2024a). However, similar to the hallucination issues in textual LLMs (Ji et al., 2023), where irrelevant content is generated with input prompts, LVLMs face a specific challenge known as object hallucination: generating non-existing objects for a given image (Li et al., 2023b; Wang et al., 2023b; Zhou et al., 2023; Fu et al., 2023; Lovenia et al., 2023; Jing et al., 2023). Such a problem is particularly concerning as it compromises the model's accuracy and reliability, especially considering the growing application of LVLMs to safety-critical downstream tasks such as medical imaging (Chambon et al., 2022; Bazi et al., 2023).

In response to the pressing issue of object hallucinations in LVLMs, early attempts (Liu et al., 2023a;b; Gunjal et al., 2023; Wang et al., 2023a) focused on addressing the bias by curating high-quality datasets for fine-tuning or leveraging advanced GPT queries (Yin et al., 2023), such as GPT-4, to post-process the generated captions. However, these methods can be infeasible to implement. For instance, creating extensive, high-quality datasets for fine-tuning LVLMs is costly and requires significant human annotation. Additionally, relying on advanced GPT models for post-processing is expensive and can raise privacy concerns, especially in sensitive fields like medical imaging. Most importantly, these approaches do not address the *intrinsic* causes of object hallucination in LVLMs. Specifically, fine-tuning simply provides more data for the LVLM to learn, which can lead to overfitting to a particular dataset, as seen with methods like LURE (Zhou et al., 2023). Post-processing methods may also introduce new hallucinations, as they do not inherently correct the root cause of hallucinations in LLMs or LVLMs but just overwrite the generated response.

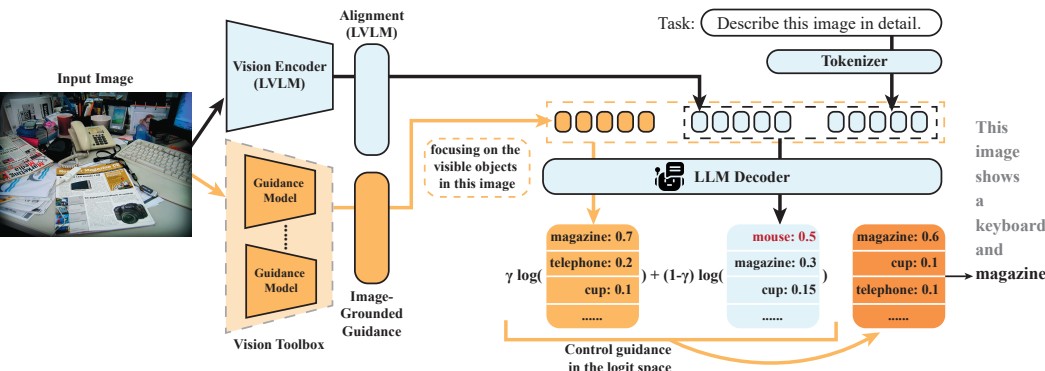

Figure 1: Illustration of MARINE framework, which introduces a vision toolbox with one or multiple guidance models to enrich the visual context of the original LVLM. The output logits are controlled to place more importance on the guided generation with the guidance strength $\gamma$.

In this paper, we investigate the intrinsic causes of object hallucination in LVLMs. Specifically, these deficiencies may stem from the three main components of the LVLMS: 1) insufficient visual context provided by the visual encoder (Zhang et al., 2023b), 2) misalignment between the vision and text domains, and 3) inherent hallucinations common in general language models. To address the first two LVLM-specific causes, we introduce **M**itigating hallucin**A**tion via image-g**R**ounded gu**I**da**N**c**E** (MARINE). MARINE mitigates hallucination issues arising from the visual encoder and domain misalignment by leveraging external guidance from image-grounded models, such as object detection models. Our approach leverages the inherent advantage of these image-grounded models, which are specifically designed and trained for more detailed visual information extraction. These models provide higher quality, fine-grained visual encoding compared to the standard visual encoders in LVLMs, which are primarily optimized for grasping the overall context of an image. Furthermore, we integrate the guidance from image-grounded models into text descriptions, allowing the LVLM to process the information without requiring additional alignment procedures. As a result, MARINE is a training-free, API-free[1] method that addresses object hallucination at inference time by targeting its two root causes.

As shown in Figure 1, MARINE incorporates one or more image-grounding models to enrich the visual context of LVLMs. The guidance are then aggregated as prompt input to the LLM decoder to improve the response quality. Empirical evaluations are conducted on five widely-recognized LVLMs across benchmarks including MSCOCO (Lin et al., 2014), LLaVA-QA90 task (Liu et al., 2023d), A-OKVQA (Schwenk et al., 2022), and GQA (Hudson & Manning, 2019). We present results based on guidance from a aggregated source of DEtection TRansformer (DETR) (Carion et al., 2020) and RAM++ (Huang et al., 2023b). We also include ideal results based on ground truth object oracle, denoted as MARINE-Truth. Our experimental results demonstrate that, in comparison with state-of-the-art algorithms, MARINE exhibits further reduced hallucination, as measured by popular hallucination metrics such as CHAIR (Rohrbach et al., 2018) and POPE (Li et al., 2023b), as well as additional metrics considered in this study including the recall and GPT-4V's evaluation of the responses. These results confirm that MARINE can effectively mitigate object hallucinations without requiring additional training resources or access to advanced LLMs. To summarize, our contribution are listed as follows:

- We introduce MARINE, a universal framework and aggregating a toolbox of image-grounded visual models to guide the generation process of LVLMs. MARINE leverages the intrinsic advantages of these visual models in providing the detailed information of the input image and help mitigate the hallucinations in LVLMs.
- Through extensive evaluations on various datasets, we demonstrate that MARINE consistently outperform the baselines in hallucination mitigation while maintaining overall performance across multiple tasks (image captioning, VQA).
- MARINE provides a favorable trade-off between latency and accuracy, with the lowest computational overhead compared to existing baselines. The minimal increase in latency comparing to the

---

[1]The term "API-free" in denotes the elimination of any need for API calls to OpenAI. We note that Woodpecker requires 3-5k input tokens for an API call to each short captioning task.

baselines, combined with the high accuracy of our results, positions `MARINE` as a practical and scalable solution for real-world applications without significant computational cost.

## 2 RELATED WORK

**Object Hallucination in Large Vision-Language Models.** Since the introduction of recent Large Vision-Language Models (LVLMs) (Liu et al., 2023d; Zhu et al., 2023; Ye et al., 2023; Dai et al., 2023a; Gao et al., 2023), the hallucination phenomenon in these models has gathered significant attention in the research community. This issue was first highlighted by Li et al. (2023b) with subsequent studies (Wang et al., 2023b; Zhou et al., 2023; Fu et al., 2023; Lovenia et al., 2023) that, LVLMs exhibit similar hallucination problems as the textual LLMs. Notably, different from textual LLMs, LVLMs are prone to a unique type of hallucination called 'object hallucination' (Rohrbach et al., 2018), where the model falsely perceives the presence of non-existent objects in images. In response to object hallucination problems, efforts have been made to mitigate object hallucination in smaller image captioning models (Biten et al., 2022; Dai et al., 2023b). Regarding the recent development of LVLMs, several works (Liu et al., 2023b; Gunjal et al., 2023) proposed vision-language fine-tuning datasets aimed for improved robustness. Wang et al. (2023a) leveraged the vision-language model to generate more diverse instruction-tuning data and iteratively correct the inaccuracies in data. Zhai et al. (2023) introduced a GPT-4 assisted evaluation method and also a fine-tuning strategy using the MSCOCO dataset. Most related to our setting, Yin et al. (2023) proposed Woodepecker, a five-stage training-free method eventually leveraging GPT-3.5 API for hallucination correction. Concurrently, several works (Leng et al., 2023; Huang et al., 2023a; Chen et al., 2024; Liu et al., 2024; Wan et al., 2024; Zhang et al., 2024b) began to focus on on the training-free setting and can similarly be formulated as approaches using Classifier-Free Guidance (CFG). However, these approaches and `MARINE` differ in the focus of their method designs within the larger framework of CFG. With detailed discussion deferred to Appendix A, we highlight that MARINE operates at the image level and uniquely employs a vision toolbox to ensemble information from multiple vision models, producing guidance that reaches a consensus among the models for more accurate results. These approaches are further complementary to `MARINE`, and integrating them could pave the way for a more effective strategy in future research.

**Controllable Generation.** Controllable text generation (Prabhumoye et al., 2020; Hu & Li, 2021; Zhang et al., 2023a) has emerged as a vital research domain, focusing on the generation of natural sentences with controllable attributes such as persona (Prabhumoye et al., 2020; Hu & Li, 2021; Zhang et al., 2023a)and politeness (Niu & Bansal, 2018; Madaan et al., 2020). Among the various approaches, fine-tuning has been recognized as the most straightforward approach, achieved either through full fine-tuning (Li & Liang, 2021; Ouyang et al., 2022; Carlsson et al., 2022) or integrating tunable adaptors (Lin et al., 2021; Ribeiro et al., 2021). While fine-tuning has been effective in a wide range of applications, it is also expensive in computation as the size of LLMs is growing tremendously. Recently, there has been a development on controllable generation with diffusion models (Li et al., 2022; Lin et al., 2023b), extending to controllable text-to-image generation (Yang et al., 2023). Particularly, the use of classifier guidance (Dhariwal & Nichol, 2021) and classifier-free guidance (Ho & Salimans, 2021) has become prominent in refining the quality of generated outputs. Most recently, Sanchez et al. (2023) applied classifier-free guidance to language models in the *single-modal* setting to improve their performance at inference time. Our approach methodologically resembles classifier-free guidance for LVLMs' text generation, while specifically addressing the *multi-modal* context and focusing on reducing hallucinations.

## 3 PRELIMINARIES

**Notation.** We use lower case letters, lower case bold face letters, and upper case bold face letters to denote scalars, vectors, and matrices respectively. We use the symbol $p$ to represent the conditional probability of LLM's response. And we denote the sequence of tokens generated before the $t$-th token as $\mathbf{y}_{<t} = [y_1, \ldots, y_{t-1}]$ for $t > 1$. $\mathbf{y}_{<t}$ is an empty sequence when $t = 1$.

**Generative language models.** Let $p_{\boldsymbol{\theta}}$ denotes an LLM parameterized by $\boldsymbol{\theta}$. Consider a sequence $\mathbf{x} = [x_1, \ldots, x_n]$ as the input prompt, where each $x_i$ is a token from a predefined vocabulary. The LLM then generates the response sequence $\mathbf{y} = [y_1, \ldots, y_m]$ by sampling from the conditional probability distribution $p_{\boldsymbol{\theta}}(\cdot|\mathbf{x})$, where $y_t$ denotes individual token for $1 \leq t \leq m$. The conditional distribution $p_{\boldsymbol{\theta}}(\mathbf{y}|\mathbf{x})$ can therefore be expressed as $p_{\boldsymbol{\theta}}(\mathbf{y}|\mathbf{x}) = \prod_{t=1}^{m} p_{\boldsymbol{\theta}}(y_t|\mathbf{x}, \mathbf{y}_{<t})$, where $\mathbf{y}_{<t} = [y_1, \ldots, y_{t-1}]$ for $t > 1$ and is empty for $t = 1$. In the case of LVLMs, visual tokens $\boldsymbol{b} = [v_1, \ldots, v_k]$

are additionally included. These tokens are generated from a pre-trained visual encoder and mapped into the token space through a linear projection. The conditional distribution of output $\mathbf{y}$ given the visual tokens $\boldsymbol{b}$ and textual prompt $\mathbf{x}$ is expressed as $p_{\boldsymbol{\theta}}(\mathbf{y}|\boldsymbol{b}, \mathbf{x}) = \prod_{t=1}^{m} p_{\boldsymbol{\theta}}(y_t|\boldsymbol{b}, \mathbf{x}, \mathbf{y}_{<t})$, where $p_{\boldsymbol{\theta}}$ is approximated by LVLMs.

**Guidance in generative models.** The process of a guided generation involves getting the output $\mathbf{y}$ conditioned on input $\mathbf{x}$, which encodes the desired properties of the output $\mathbf{y}$. This guidance can be generally added to the model by two distinct approaches: classifier guidance (Dhariwal & Nichol, 2021) and classifier-free guidance (Ho & Salimans, 2021). As a top-level view, both methods formulate the conditional probability distribution of output $\mathbf{y}$ conditioned on guidance $\mathbf{x}$ as

$$p(\mathbf{y}|\mathbf{x}) \propto p_{\boldsymbol{\theta}}(\mathbf{y})p(\mathbf{x}|\mathbf{y})^{\gamma}, \tag{3.1}$$

where $p_{\boldsymbol{\theta}}(\mathbf{y})$ is the original generative model and $p(\mathbf{x}|\mathbf{y})$ is the posterior distribution of $\mathbf{x}$ given $\mathbf{y}$ and $\gamma$ is the guidance strength. In the classifier guidance, the posterior distribution $p(\mathbf{x}|\mathbf{y})$ in equation 3.1 is replaced by a classifier $p_{\phi}(\mathbf{x}|\mathbf{y})$ parameterized by $\phi$, which requires additional training step and calculating $\nabla_{\mathbf{x}} \log p_{\phi}(\mathbf{x}|\mathbf{y})$. The classifier-free guidance, on the other hand, removes the necessity of the parameterized classifier $f_{\phi}$. Instead, according to the Bayes rule, the posterior distribution can be approximated by $p_{\boldsymbol{\theta}}(\mathbf{x}|\mathbf{y}) \propto p_{\boldsymbol{\theta}}(\mathbf{y}|\mathbf{x})/p_{\boldsymbol{\theta}}(\mathbf{y})$, where $p_{\boldsymbol{\theta}}(\mathbf{y}|\mathbf{x})$ is the generative model when taking $\mathbf{x}$ as prompt input. Plugging this back into equation 3.1 yields the guided distribution that can be approximated by

$$\widehat{p}_{\boldsymbol{\theta}}(\mathbf{y}|\mathbf{x}) \propto p_{\boldsymbol{\theta}}(\mathbf{y}) \cdot p_{\boldsymbol{\theta}}(\mathbf{y}|\mathbf{x})^{\gamma}/p_{\boldsymbol{\theta}}(\mathbf{y})^{\gamma} = p_{\boldsymbol{\theta}}(\mathbf{y}|\mathbf{x})^{\gamma}/p_{\boldsymbol{\theta}}(\mathbf{y})^{\gamma-1}.$$

As a result, the guided LLM $\widehat{p}_{\boldsymbol{\theta}}$ places more importance on the prompt $\mathbf{x}$ during generation with the increasing value of $\gamma$, thereby producing texts that better align with the desired behavior from the prompt (Sanchez et al., 2023).

## 4 METHOD

The existing architecture of LVLMs is usually composed of a visual encoder, a visual and textual domain alignment layer, and the LLM itself. Therefore, besides the inherent language priors of LLMs (Biten et al., 2022), object hallucination may arise from (1) deficiencies in the visual encoder providing insufficient visual information (Zhang et al., 2023b) and (2) misalignment between the visual and textual domains. To mitigate object hallucinations, we introduce MARINE, a framework containing two major components to address the aforementioned challenges: (1) introducing additional visual information from a set of vision models and (2) using the additional aggregated visual features to guide the LVLM's generation. In Figure 1, we present the framework overview.

### 4.1 VISUAL GUIDANCE FROM IMAGE-GROUNDED FEATURES

To introduce image-grounded guidance to mitigate hallucinations, our approach integrates additional object detection models, which differ from the visual encoders used in LVLM that are usually pre-trained from CLIP (Radford et al., 2021). This integration leverages the object detection models to extract detailed visual information from images. Upon acquiring these extra visual information from different image-grounded models, we aggregate and translate the collected information into textual information. This aggregation can be done by the language model (Lin et al., 2023a) or rule based algorithm (Bird et al., 2009). Such an information aggregation is effective and efficient, as it eliminates the necessity of fine-tuning the alignment layer while retaining the rich information encoded by various of image grounding models. We subsequently employ a simple prompt "focusing on the visible objects in this image:" and concatenate it with the aggregated object information, denoted as the guidance prompt $\mathbf{c}$.

### 4.2 GUIDED TEXT GENERATION WITH VISUAL INFORMATION

We tackle the object hallucination problem of LVLMs by specifically placing importance on the addtional image-grounded information we introduced. In addition to the visual tokens $\boldsymbol{b}$ extracted from the original LVLM and textual prompt $\mathbf{x}$, we extract the auxiliary visual tokens $\mathbf{c}$ from the additional guidance models. The generation of the $t$-th token in the output $\mathbf{y}$ of our classifier-free guided LVLM $p_{\boldsymbol{\theta}}$ is expressed as

$$\widehat{p}_{\boldsymbol{\theta}}(y_t|\boldsymbol{b}, \mathbf{c}, \mathbf{x}, \mathbf{y}_{<t}) \propto p_{\boldsymbol{\theta}}(y_t|\boldsymbol{b}, \mathbf{c}, \mathbf{x}, \mathbf{y}_{<t})^{\gamma}/p_{\boldsymbol{\theta}}(y_t|\boldsymbol{b}, \mathbf{x}, \mathbf{y}_{<t})^{\gamma-1},$$

where $\mathbf{c}$ denotes our control guidance and $\gamma$ is the control strength. The sampling of output generation is given by

$$\widehat{p}_{\boldsymbol{\theta}}(\mathbf{y}|\boldsymbol{b},\mathbf{c},\mathbf{x}) = \prod_{t=1}^{m}\widehat{p}_{\boldsymbol{\theta}}(y_t|\boldsymbol{b},\mathbf{c},\mathbf{x},\mathbf{y}_{<t}) \propto \prod_{t=1}^{m}\frac{p_{\boldsymbol{\theta}}(y_t|\boldsymbol{b},\mathbf{c},\mathbf{x},\mathbf{y}_{<t})^{\gamma}}{p_{\boldsymbol{\theta}}(y_t|\boldsymbol{b},\mathbf{x},\mathbf{y}_{<t})^{\gamma-1}} = \frac{p_{\boldsymbol{\theta}}(\mathbf{y}|\boldsymbol{b},\mathbf{c},\mathbf{x})^{\gamma}}{p_{\boldsymbol{\theta}}(\mathbf{y}|\boldsymbol{b},\mathbf{x})^{\gamma-1}}.$$

We can further view MARINE in the logit space, where the $t$-th token is therefore sampled from the logit space by

$$\log\widehat{p}_{\boldsymbol{\theta}}(y_t|\boldsymbol{b},\mathbf{c},\mathbf{x},\mathbf{y}_{<t}) = \gamma\log p_{\boldsymbol{\theta}}(\mathbf{y}|\boldsymbol{b},\mathbf{c},\mathbf{x},\mathbf{y}_{<t}) + (1-\gamma)\log p_{\boldsymbol{\theta}}(\mathbf{y}|\boldsymbol{b},\mathbf{x},\mathbf{y}_{<t}).$$

This linear combination of logits implies that the conditional generation on the additional image-grounded guidance acts as a controllable gate. Only objects with relatively high probabilities in both branches could appear at top when sampling. Specifically, setting $\gamma = 0$ recovers the original LLM generation without control guidance and setting $\gamma = 1$ produces the LLM generation entirely based on the control. Meanwhile, for $\gamma \in (0, 1)$, MARINE yields a combination of the original generation $p_{\boldsymbol{\theta}}(\mathbf{y}|\boldsymbol{b},\mathbf{x})$ and the generation conditioned on the guidance $p_{\boldsymbol{\theta}}(\mathbf{y}|\boldsymbol{b},\mathbf{c},\mathbf{x})$. This strikes a balance between a better ability to follow instructions to generate high-quality answers and the increased accuracy and detail in image descriptions. The formulation therefore shares resemblance to the classifier-free guidance introduced for LLMs (Sanchez et al., 2023), which places importance on the textual prompt itself to better align the LLM generation with user intention in the *single-modal* setting. We summarize MARINE in Algorithm 1. In detail, MARINE aggregates the collected visual information $\{\mathbf{c}_i\}_i$ using function Aggr., which can be a small language model for information aggregation (Lin et al., 2023a), or a rule-based algorithms like majority voting (as similarly used by Wang et al.). Notably, MARINE only double the LLM inference time of in Line 7 and Line 9, while adding the guidance from each single image grounded model will significantly increase the inference time when the number of image grounded models increase.

---

**Algorithm 1** **M**itigating hallucin**A**tion via image-g**R**ounded gu**I**da**N**c**E** (MARINE)

---
1: **Input:** LLM parameter $\boldsymbol{\theta}$, input prompt $\mathbf{x}$, visual tokens $\boldsymbol{b}$ from LVLM's original vision tower
2: **Input:** auxiliary visual tokens $\{\mathbf{c}_i\}_{i=1}^{M}$ from $M$ image grounding models, guidance scale $\gamma$
3: Initialize empty output $\mathbf{y} = []$.
4: Aggregate visual information as textual prompt $\mathbf{c} = \text{Aggr.}(\{\mathbf{c}_i\}_{i=1}^{M})$
5: **for** $t = 0, 1, \ldots, T$ **do**
6:     Construct unconditional input $\mathbf{x}_{\text{uncond}}^{(t)} = [\boldsymbol{b}, \mathbf{x}, \mathbf{y}_{<t}]$.
7:     Generate unconditional output logits using LLM: $\ell_{\text{uncond}}^{(t)} = \log p_{\boldsymbol{\theta}}(\mathbf{x}_{\text{uncond}}^{(t)})$.
8:     Construct conditional input $\mathbf{x}_{\text{cond}}^{(t)} = [\boldsymbol{b}, \mathbf{c}, \mathbf{x}, \mathbf{y}_{<t}]$.
9:     Generate conditional output logits using LLM: $\ell_{\text{cond}}^{(t)} = \log p_{\boldsymbol{\theta}}(\mathbf{x}_{\text{cond}}^{(t)})$.
10:    Update output logits $\ell^{(t)} = \gamma\ell_{\text{cond}}^{(t)} + (1-\gamma)\ell_{\text{uncond}}^{(t)}$.
11:    Sample token $y_t$ from logit space denoted by $\ell^{(t)}$.
12:    Let $\mathbf{y} = [\mathbf{y}, y_t]$.
13: **end for**
14: **Output:** $\mathbf{y}$.

---

## 5 EXPERIMENTS

In this section, we evaluate MARINE in mitigating object hallucinations across various LVLMs, showing that it outperforms state-of-the-art methods on established metrics across different question formats.

### 5.1 EXPERIMENT SETUP

**Models.** To demonstrate the broad applicability of our approach across different LVLM architectures, we apply and evaluate MARINE to recent widely-used models including *LLaVA* (Liu et al., 2023d), *LLaVA-v1.5* (Liu et al., 2023c), *MiniGPT-v2* (Chen et al., 2023), *mPLUG-Owl2* (Ye et al., 2023) and *InstructBLIP* (Liu et al., 2023c). To address the object hallucination problems in text generation, we incorporate the DEtection TRansformer (DETR) (Carion et al., 2020) and RAM++ (Huang et al., 2023b) as the additional vision models for guidance.

**Guidance from Multiple Sources.** Our framework's compatibility with various vision models allows for the incorporation of multiple sources to enhance precision and robustness. By considering

object-level information from DETR and RAM++ simultaneously, we generate guidance that reflects consensus across these models. This approach significantly improves the accuracy and reliability of the guidance provided to the LVLM.

**Datasets and evaluations.** In alignment with established evaluations from previous studies (Dai et al., 2023b; Yin et al., 2023), we assess our method using the following metrics:

- Caption Hallucination Assessment with Image Relevance (*CHAIR*) (Rohrbach et al., 2018). It involves prompting the LVLMs to generate a description for the input image, and then comparing this generation with ground truth objects present in the image. CHAIR quantifies hallucination both at instance level and sentence level, respectively defined as $\text{CHAIR}_I$ and $\text{CHAIR}_S$:

$$\text{CHAIR}_I = \frac{\left|\{\text{hallucinated objects}\}\right|}{\left|\{\text{all mentioned objects}\}\right|}, \quad \text{CHAIR}_S = \frac{\left|\{\text{captions with hallucinated objects}\}\right|}{\left|\{\text{all captions}\}\right|}.$$

In addition to these metrics, we incorporate an instance-level Recall score in our evaluation to evaluate whether the descriptions accurately include the necessary visual content from the image:

$$\text{Recall} = \left|\{\text{non-hallucinated objects}\}\right| / \left|\{\text{all existing objects}\}\right|.$$

- Polling-based Object Probing Evaluation (*POPE*) (Li et al., 2023b). POPE formulates a binary classification task by prompting LVLMs with questions such as "Is there a keyboard in this image?" to answer "yes" or "no". We specifically focus on the adversarial setting, which is considered the most challenging setting. Results for the random and popular settings are detailed in Appendix E. We report the accuracy and F1 score of the LVLMs' responses, and the proportion of "yes" answers.
- *GPT-4V-aided Evaluation* (Yin et al., 2023). The GPT-4V-aided evaluation compares the outputs of two LVLM assistants using GPT-4V as a judge. In this evaluation, we utilize the LLaVA-QA90 task (Liu et al., 2023d)[2] (including conversations, visual perceptions, and complex reasoning tasks) and additionally consider the image captioning task.

Consistent with Li et al. (2023b), we randomly sampled a subset of 500 images from MSCOCO (Lin et al., 2014) dataset for CHAIR evaluation. For the POPE evaluation, we created 3000 questions across three datasets—500 images each from MSCOCO, A-OKVQA (Schwenk et al., 2022), and GQA (Hudson & Manning, 2019). For the GPT-4V-aided evaluation, we utilized 90 questions from the LLaVA-QA90 task and randomly selected 50 MSCOCO images for image captioning task.

**Baselines.** In addition to comparing with the performance of the original LVLM sampling method, we also consider the following popular methods for mitigating hallucinations.

- *Greedy-Decoding*, which adopts the greedy sampling strategy, by generating tokens with the highest posterior probability to address hallucinations arising from.
- *LURE* (Zhou et al., 2023), which identifies and masks potentially hallucinated words and fine-tune a MiniGPT4 model to rectify object hallucinations in the generated descriptions.
- *LURE with Cutoff*. The original LURE method tends to generate long descriptions regardless of the provided instructions, which sometimes results in even worse performance as unnecessary information is included. Therefore, we also introduce a modified baseline, where we truncate the LURE's output to match the length (in terms of the number of sentences) of the original generations.
- *Woodpecker* (Yin et al., 2023), which leverages GPT-3.5 to correct hallucinations in LVLM generation with five steps toward the correction.
- *VCD* (Leng et al., 2023), which distorts the image inputs to impose penalties on logit outputs.
- *OPERA* (Huang et al., 2023a), which penalizes logits to mitigate over-trust in beam-search decoding and adjusts token selection.

Lastly, the performance of MARINE improves in correlation with the advancement of the control guidance extractor used. Consequently, to demonstrate the potential upper bound of MARINE's performance, we consider a version utilizing a ground-truth oracle extractor, which we denote as MARINE-Truth. Further details on model architectures, datasets and evaluation metrics are deferred to Appendix C.

**Hyperparameter Setting.** The hyperparameters for our method are fixed across tasks, with key settings including a guidance strength of 0.7, noise intensity for DETR at 0.95, a detection threshold for RAM++ of 0.68, and a greedy sampling approach with a random seed of 242.

---

[2] https://github.com/haotian-liu/LLaVA/blob/main/playground/data/coco2014_val_gpt4_qa_30x3.jsonl

Table 1: Evaluation with CHAIR score across multiple LVLM architectures comparing our method with several baselines. We report CHAIR$_S$, CHAIR$_I$ and the recall score. The **bold** numbers indicate the best results among the methods evaluated and the underscored numbers represent the second-best results. We show MARINE-Truth as a reference performance of MARINE.

| Method | LLaVA | | | LLaVA-v1.5 | | | MiniGPTv2 | | | mPLUG-Owl2 | | | InstructBLIP | | | Average | | |
|---|---|---|---|---|---|---|---|---|---|---|---|---|---|---|---|---|---|---|
| **CHAIR** | $C_S\downarrow$ | $C_I\downarrow$ | $R\uparrow$ | $C_S\downarrow$ | $C_I\downarrow$ | $R\uparrow$ | $C_S\downarrow$ | $C_I\downarrow$ | $R\uparrow$ | $C_S\downarrow$ | $C_I\downarrow$ | $R\uparrow$ | $C_S\downarrow$ | $C_I\downarrow$ | $R\uparrow$ | $C_S\downarrow$ | $C_I\downarrow$ | $R\uparrow$ |
| Greedy | 26.6 | 10.5 | 47.4 | 8.8 | 4.6 | 41.1 | 8.2 | 4.2 | 41.1 | 6.2 | 3.4 | 38.8 | 5.0 | 3.2 | 33.2 | 11.0 | 5.2 | 40.3 |
| LURE | 33.8 | 11.6 | **54.8** | 38.9 | 11.2 | **56.3** | 36.2 | 11.4 | **54.6** | 33.9 | 10.8 | **55.9** | 38.1 | 12.1 | **54.5** | 36.2 | 11.4 | **55.2** |
| LURE w/ cutoff | 24.4 | 9.3 | 50.2 | 18.4 | 6.8 | 47.3 | 12.5 | 6.2 | 42.0 | 15.4 | 6.6 | 45.5 | 9.6 | 6.4 | 34.5 | 16.1 | 7.1 | 43.9 |
| Woodpecker | 19.5 | 8.9 | 44.3 | 8.5 | 4.5 | 38.4 | 7.5 | 4.5 | 37.0 | 8.0 | 4.3 | 37.5 | 8.0 | 6.2 | 32.6 | 10.3 | 5.7 | 38.0 |
| VCD | 28.1 | 11.0 | 46.6 | 7.3 | 4.1 | 40.8 | 6.8 | 3.9 | 38.2 | 5.9 | 3.4 | 37.7 | 2.4 | 1.5 | 33.7 | 10.1 | 4.8 | 39.4 |
| OPERA | 22.4 | 9.9 | 43.6 | 11.0 | 6.7 | 40.2 | 9.2 | 5.0 | 41.3 | 5.8 | 3.2 | 38.4 | 4.6 | 2.7 | 38.0 | 10.6 | 5.5 | 40.3 |
| **MARINE** | **17.8** | **7.2** | 50.8 | **6.2** | **3.0** | 44.3 | 11.8 | 4.9 | 49.7 | **4.2** | **2.3** | 41.4 | **2.2** | **1.3** | 36.3 | **8.4** | **3.7** | 44.5 |
| MARINE-Truth | 19.6 | 5.1 | 79.0 | 6.0 | 2.5 | 55.3 | 12.6 | 3.8 | 70.5 | 3.8 | 1.7 | 48.0 | 3.0 | 1.8 | 35.9 | 8.9 | 2.9 | 57.5 |

Table 2: Evaluation with POPE score in adversarial setting across multiple LVLM architectures comparing our method with several baselines. We report the POPE accuracy (%), F1 score (%) and the yes ratio (%). The ideal yes ratio for a non-biased LVLM is 50%. The **bold** numbers indicate the best results among the methods evaluated and the underscored numbers represent the second-best results. We show MARINE-Truth as a reference performance of MARINE.

| Method | LLaVA | | | LLaVA-v1.5 | | | MiniGPTv2 | | | mPLUG-Owl2 | | | InstructBLIP | | | Average | | |
|---|---|---|---|---|---|---|---|---|---|---|---|---|---|---|---|---|---|---|
| **POPE** | Acc$\uparrow$ | F1$\uparrow$ | Yes | Acc$\uparrow$ | F1$\uparrow$ | Yes | Acc$\uparrow$ | F1$\uparrow$ | Yes | Acc$\uparrow$ | F1$\uparrow$ | Yes | Acc$\uparrow$ | F1$\uparrow$ | Yes | Acc$\uparrow$ | F1$\uparrow$ | Yes |
| Greedy | 51.8 | 67.4 | 97.7 | 79.4 | 81.6 | 61.6 | 82.7 | 81.7 | 44.5 | 72.5 | 77.5 | 72.4 | 79.8 | 81.4 | 58.6 | 73.2 | 77.9 | 67.0 |
| LURE | - | - | - | - | - | - | - | - | - | - | - | - | - | - | - | - | - | - |
| Woodpecker | **77.5** | 77.6 | 50.5 | 80.5 | 80.6 | 50.5 | 79.5 | 77.8 | 42.5 | 77.5 | 76.9 | 47.5 | 79.0 | 78.6 | 48.0 | 78.8 | 78.3 | 47.8 |
| VCD | 54.6 | 68.5 | 94.0 | 78.2 | 80.7 | 62.8 | 81.4 | 80.2 | 44.1 | 72.3 | 77.0 | 70.5 | 79.7 | 80.9 | 56.7 | 73.2 | 77.5 | 65.6 |
| OPERA | 51.7 | 67.4 | 98.0 | 77.5 | 80.1 | 63.2 | 82.9 | 81.9 | 44.3 | 70.3 | 79.1 | 84.6 | 79.8 | 81.4 | 58.6 | 72.4 | 78.0 | 69.7 |
| **MARINE** | 66.9 | 72.9 | 72.3 | **85.0** | **84.3** | 45.7 | **83.0** | **82.9** | **49.4** | **82.8** | **82.7** | **49.2** | **81.7** | 79.4 | 38.8 | **79.9** | **80.4** | **51.1** |
| MARINE-Truth | 75.6 | 80.1 | 72.3 | 92.0 | 92.5 | 57.0 | 86.9 | 88.3 | 62.5 | 93.4 | 93.76 | 56.2 | 93.8 | 93.8 | 51.0 | 88.3 | 89.7 | 59.8 |

## 5.2 RESULTS

Experimental results on object hallucination metrics (CHAIR and POPE) are presented in Table 1 and Table 2. Overall, MARINE achieves superior performances across different LVLM architectures and evaluation metrics, ranked as the best or second-best on the majority of the evaluation metrics.

**Results on CHAIR.** Table 1 presents the evaluation of various mitigation methods using CHAIR scores across multiple LVLM architectures. The results demonstrate that MARINE consistently outperforms other state-of-the-art methods, achieving the highest average scores in both CHAIR$_S$ and CHAIR$_I$ and the second-best Recall score. Specifically, MARINE surpasses the second-best performing method by an average margin of 1.7 on CHAIR$_S$ and 1.1 on CHAIR$_I$. Notably, MARINE exhibits exceptional performance on LLaVA architectures, with improvements in CHAIR scores of up to 8.8 compared to its original performance. In contrast, methods such as LURE and Woodpecker show less effectiveness in hallucination mitigation. The reference method, MARINE-Truth, generally achieves the strongest results, as expected given its access to ground-truth guidance. However, MARINE's performance closely approximates that of its ground-truth counterpart, indicating successful leveraging of multiple guidance models to provide reliable control in LVLM generation.

**Results on POPE.** The POPE evaluation, presented in Table 2, further validates the superior performance of MARINE against existing baselines across various question formats. MARINE consistently outperforms all other methods by a substantial margin, demonstrating average improvements of 6.7% in accuracy and 3.5% in F1 score relative to the original outputs across models. Even when compared to the second-best method, Woodpecker, MARINE maintains a performance edge of 1.1% and 2.1% respectively in accuracy and F1 score. Moreover, MARINE effectively mitigates the LVLMs' biased tendency towards affirmative responses, as evidenced by a more balanced "yes" ratio (closer to 50%, representing a 15.9% shift towards unbiased answers). This improvement notably addresses the overconfidence issue prevalent in existing models.

**Results on GPT-4V-aided evaluation.** Following Yin et al. (2023), we leverage GPT-4V[3] to evaluate and compare the performance of the original LVLMs and LVLMs with MARINE on LLaVA-QA90 and an image captioning task. This GPT-4V-assisted evaluation introduces a qualitative

---

[3]We used gpt-4-1106-vision-preview in obtaining our final experiment results. As OpenAI continues to update its API, different versions may result in slightly different values.

Table 3: Results of GPT-4V-aided evaluation. The accuracy and detailedness metrics are on a scale of 10, and a higher score indicates better performance. The symbols $\times$ and $\checkmark$ indicate performance metrics without and with our method, respectively.

| Task | Metrics | LLaVA | | mPLUG-Owl2 | |
|---|---|---|---|---|---|
| | | $\times$ | $\checkmark$ | $\times$ | $\checkmark$ |
| LLaVA-QA90 | Acc ↑ | $5.82_{\pm0.10}$ | $\mathbf{5.94}_{\pm0.05}$ | $6.03_{\pm0.13}$ | $\mathbf{6.35}_{\pm0.21}$ |
| | Detail ↑ | $4.59_{\pm0.08}$ | $4.59_{\pm0.08}$ | $5.06_{\pm0.05}$ | $\mathbf{5.16}_{\pm0.10}$ |
| Image Captioning | Acc ↑ | $5.27_{\pm0.20}$ | $\mathbf{6.11}_{\pm0.23}$ | $7.97_{\pm0.25}$ | $\mathbf{8.63}_{\pm0.20}$ |
| | Detail ↑ | $\mathbf{4.39}_{\pm0.29}$ | $4.36_{\pm0.17}$ | $5.74_{\pm0.24}$ | $\mathbf{6.19}_{\pm0.23}$ |

Table 4: POPE results across three datasets. We report the average score under random, popular, adversarial settings. The detailed POPE results can be found in the appendix E. The **bold** numbers indicate the best results. The ideal yes ratio for a non-biased LVLM is 50%.

| Dataset | w/MARINE | LLaVA | | | mPLUG-Owl2 | | |
|---|---|---|---|---|---|---|---|
| | | Accuracy ↑ | F1 ↑ | Yes(%) | Accuracy ↑ | F1 ↑ | Yes(%) |
| MSCOCO (Lin et al., 2014) | $\times$ | 54.2 | 68.5 | 95.5 | 76.7 | 80.4 | 68.2 |
| | $\checkmark$ | **72.2** | **76.4** | **66.9** | **85.5** | **85.0** | **46.5** |
| A-OKVQA (Schwenk et al., 2022) | $\times$ | 51.8 | 67.5 | 97.9 | 69.6 | 76.5 | 78.5 |
| | $\checkmark$ | **64.3** | **72.8** | **80.2** | **82.0** | **83.5** | **57.2** |
| GQA (Hudson & Manning, 2019) | $\times$ | 52.0 | 67.6 | 97.8 | 73.7 | 78.7 | 72.6 |
| | $\checkmark$ | **62.5** | **71.8** | **81.8** | **80.1** | **80.6** | **51.1** |

perspective beyond the numerical metrics of CHAIR and POPE, offering a richer assessment of model performance. The evaluation prompt is detailed in Appendix C.5. As shown in Table 3, GPT-4V consistently assigns higher accuracy with equal detailedness scores to models enhanced by MARINE, highlighting its ability to produce more precise and detailed descriptions, which demonstrates the robustness of our method in real-world visual tasks.

**Additional Results on Other Vision-Language Tasks.** To further evaluate the generalizability of our approach beyond object hallucination and the MSCOCO dataset, we extended our evaluations to additional datasets including A-OKVQA and GQA and included more general caption quality metrics. As shown in Table 4, the POPE results on datasets such as MSCOCO, A-OKVQA, and GQA demonstrate that our method consistently mitigates hallucinations across various datasets with different image distributions. Figure 2 presents a comprehensive evaluation of the image captioning task on MSCOCO and LLaVA-QA90, a comprehensive VQA dataset, using metrics including BLEU(Papineni et al., 2002), ROUGE(Lin, 2004), CIDEr(Vedantam et al., 2015) and SPICE(Anderson et al., 2016). These results demonstrate that, although our method primarily targets hallucination mitigation, it maintains the overall performance of LVLMs on broader tasks, with no significant trade-offs in caption or VQA quality.

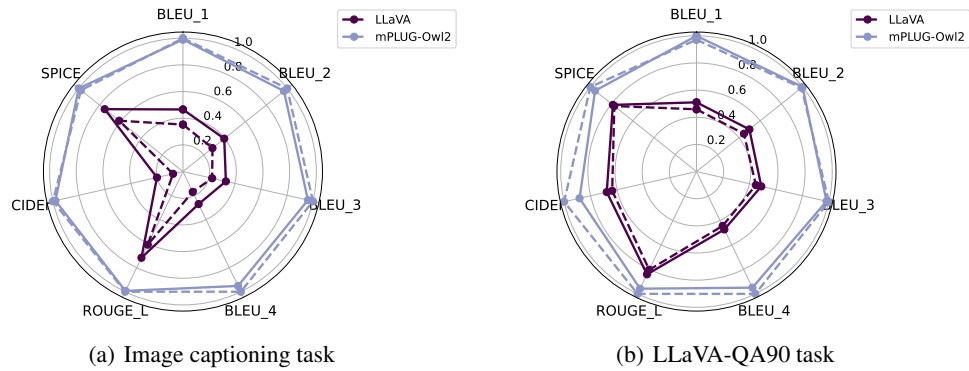

(a) Image captioning task          (b) LLaVA-QA90 task

Figure 2: MARINE leads to consistent enhancement in the text qualities on general metrics. Dashed lines and solid lines represent without or with MARINE. Higher scores indicate better quality and greater similarity between the generated captions and the reference texts.

Table 5: Inference Latency Comparison. We report both the latency and the ratio to the latency of greedy decoding of the original LVLM model.

|  | Greedy | LURE | Woodpecker* | VCD | OPERA | **MARINE** (ours) |
|---|---|---|---|---|---|---|
| Training Cost | 0 | 10min on A100 80G | 0 | 0 | 0 | 0 |
| Inference Latency$^{(ms/token)}$ | 26.3 (×1.0) | 179.9 (×6.84) | 94.5 (×3.59)* | 53.4 (×2.03) | 185.1 (×7.0) | **52.2 (×1.98)** |

*Woodpecker requires GPT API key access and the latency may depend on OPENAI API.

Table 6: Ablation study comparing the performance of combining DETR and RAM++ models versus using individual vision models. This approach leverages multiple object detectors to provide more reliable and robust object-level guidance, resulting in superior performance on CHAIR metrics.

| Model | LLaVA | | LLaVA-v1.5 | | mPLUG-Owl2 | |
|---|---|---|---|---|---|---|
| CHAIR | $C_S\downarrow$ | $C_I\downarrow$ | $C_S\downarrow$ | $C_I\downarrow$ | $C_S\downarrow$ | $C_I\downarrow$ |
| *Ensembling Models* | | | | | | |
| MARINE | **17.8** | **7.2** | **6.2** | **3.0** | **4.2** | **2.3** |
| *Single Models* | | | | | | |
| MARINE-DETR only | 27.6 | 8.4 | 10.5 | 4.3 | 5.3 | 2.7 |
| MARINE-RAM only | 29.0 | 9.1 | 6.6 | 3.7 | 5.2 | 2.8 |

Table 7: Effect of Integration Methods for Image-Grounding Models.

| Model | LLaVA | | LLaVA-v1.5 | | mPLUG-Owl2 | |
|---|---|---|---|---|---|---|
| CHAIR | $C_S\downarrow$ | $C_I\downarrow$ | $C_S\downarrow$ | $C_I\downarrow$ | $C_S\downarrow$ | $C_I\downarrow$ |
| MARINE-intersection (ours) | **17.8** | **7.2** | 6.2 | 3.0 | **4.2** | **2.3** |
| MARINE-union | 30.4 | 9.7 | **5.4** | **2.7** | 4.8 | 2.7 |

**Latency Analysis** Mitigating object hallucination often requires additional computational resources, a characteristic common to many existing methods which typically involve additional post-generation correction models (Zhou et al., 2023; Zhai et al., 2023; Yin et al., 2023), object detectors (Yin et al., 2023), or more complex decoding processes (Huang et al., 2023a; Leng et al., 2023) to reduce hallucinations. Furthermore, to assess the practical feasibility of our approach in terms of computational costs, we have compared our method with existing baselines on LLaVA-7B. As demonstrated in Table 5, our method increases the decoding time by factors of 1.98, which is the lowest costs among existing baselines, suggesting MARINE can be widely applied with negligible cost. Our method offers the most favorable trade-offs between latency and accuracy in hallucination mitigation. Detailed experiment setting is in Appendix C.6.

## 5.3 ABLATION STUDY

**How Does Incorporating Multiple Sources to Form Guidance Impact Performance?** We perform an ablation study to assess the impact of incorporating DETR and RAM++ compared to using each model individually, as presented in Table 6. Notably, DETR allows for highly accurate object detection, while RAM++ excels in extensive recognition tasks, adding fine-grained details to image understanding. Combining the strengths of these image-grounding models, we achieve significant performance improvements on the CHAIR metrics. This demonstrates that leveraging complementary visual contexts can substantially enhance overall model effectiveness.

**Which Method of Integrating Image-Grounding Models Works Best?** We investigate two approaches for integrating image-grounding models: using either the intersection or union of detected objects. As shown in Table 7, the intersection-based method outperforms the union, significantly reducing object hallucination. This result highlights the importance of precision and consistency in guidance, as taking intersection ensures consensus across models, leading to more reliable guidance. The detailed experimental setup and prompt templates are provided in Appendix C.

**Effect of Guidance Strength.** Figure 3 shows that increasing guidance strength from 0 to 1 leads to a notable decrease in CHAIR scores. This trend suggests that higher guidance strength makes LVLMs rely more on image-grounded features provided by DETR, thereby enhancing their ability to produce accurate descriptions. It's crucial to note that, although some models exhibit optimal performance at a guidance strength of $\gamma = 1$, excessively strong guidance can adversely affect the models' ability to adhere to provided instructions. Experimental evidence is detailed in Appendix E. This observation highlights the necessity of having a balanced guidance strength that ensures high-quality, accurate outputs while adhering closely to the given instructions. Based on our findings, we recommend

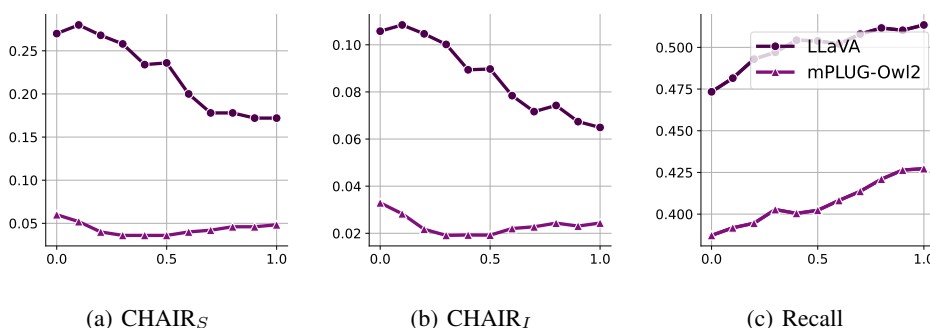

(a) CHAIR$_S$         (b) CHAIR$_I$         (c) Recall

Figure 3: Ablation study on the effect of guidance strength ($\gamma$) on the performance of LLaVA and mPLUG-Owl2 using CHAIR metrics, with $\gamma$ ranging from 0 to 1.

a guidance strength within the range of $\gamma \in (0.3, 0.7)$ as the most effective for maintaining this balance.

**Examples of `MARINE`'s Guided Generation.** In Figure 4, we provide specific generation examples of LLaVA based on queries from different tasks, with or without `MARINE`. In the first example, LLaVA incorrectly identifies a white chair in the image, an instance of hallucination as the object present is a white bird instead. In contrast, `MARINE` successfully mitigates this hallucination, correctly guiding the the model to recognize the object as a white bird. Similarly, in the second example, LLaVA erroneously state that the skateboard rider is holding onto the trucks. With `MARINE`, the model's response is more accurate, focusing on verifiable visual elements and correctly stating that the person is standing on the skateboard without introducing non-existent details.

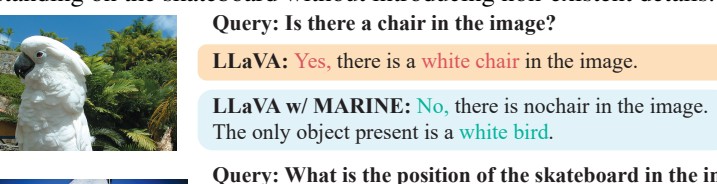

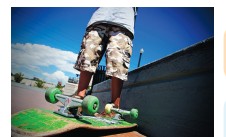

Figure 4: Examples of hallucination mitigation by our proposed `MARINE` across multiple tasks: POPE on the GQA dataset, LLaVA-QA90 task on the MSCOCO dataset, and image captioning. Hallucinated objects generated by LVLM are highlighted in red.

## 6 CONCLUSIONS, LIMITATIONS AND FUTURE WORK

In this paper, we introduced a training-free and API-free framework `MARINE` to mitigate object hallucination in LVLMs during its text generation process. Leveraging a pre-trained object grounding vision encoder for a novel guidance framework in the multi-modal setting, `MARINE` effectively and cost-efficiently reduces the hallucinations of five widely-used LVLMs, as assessed by various metrics across different tasks. The inherent compatibility of the `MARINE` with various vision models and projection functions further underscores its flexibility. In contrast to post-generation correction methods, `MARINE` strikes a balance between efficiency, instruction-following ability and effectiveness in reducing object hallucinations.

**Limitations and future work.** While `MARINE` has demonstrated impressive performance by utilizing guidance from image-grounded models, there remains potential for further improvement through the integration of advanced aggregation methods, such as multi-agent debate (Du et al., 2023), into the `MARINE` framework. Additionally, although `MARINE` is specifically designed to mitigate object hallucination, which is the most significant issue in LVLMs, extending its application to address other types of hallucinations in both LLMs and LVLMs across a broader range of benchmarks would be highly advantageous.

## ETHICS STATEMENT

This paper introduces research aimed at advancing the field of Large Language Models. We are confident that our work will contribute to significant social benefits, particularly by enhancing the accountability of LLMs through the reduction of hallucinatory outputs. Our proposed method, `MARINE`, holds the potential to improve the fairness of LLM interactions by effectively reducing biased hallucinations. To the best of our knowledge, we have not identified any negative effects associated with our research that merit highlighting in this discussion.

## REPRODUCIBILITY STATEMENT

We provide detailed descriptions of our experimental setups, datasets, models, codes in the supplementary materials to ensure the reproducibility of `MARINE`. The full experimental settings and hyperparameters are presented in Appendix C.

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

## A  BROADER IMPACT STATEMENT

By mitigating hallucinations, MARINE has the potential to offer a positive social impact by ensuring that LVLMs generate more accountable responses. Despite this merit, MARINE cannot address prejudicial biases inherent in LLM prior knowledge, which could be a focus of future work. Aside from this limitation, we do not foresee any potential negative impacts of our work that need to be specifically highlighted at this time, to the best of our knowledge.

## B  CONCURRENT WORKS.

Concurrently, Leng et al. (2023) introduced Visual Contrastive Decoding (VCD), a technique that applies noise to image inputs and penalizes logit outputs of these corrupted images. Huang et al. (2023a) enhanced beam-search decoding with the Over-trust Penalty and Retrospection-Allocation Strategy (OPERA), which penalizes over-trust and refines token selection based on previous outputs. HALC (Chen et al., 2024) employs adaptive focal-contrast decoding to encourage LVLMs to focus on fine-grained visual information, while using a computationally intensive beam search algorithm. Liu et al. (2024) addresses the issue of "Text Inertia", where LVLMs produce the same hallucinated object descriptions regardless of whether an image is actually provided. Wan et al. (2024) focuses on fine-grained guidance and especially operates at the sub-image level. Zhang et al. (2024b) adjusts the attention weights of the visual tokens within the attention mechanism to control the focus of the generation on the highlighted parts.

## C  EXPERIMENT DETAILS

We conduct all of the experiments using 8 A6000 GPU with 48GB GPU memory. Each single experiment can be run on a single A6000 GPU.

### C.1  MODEL ARCHITECTURES

In Table 8, we provide detailed descriptions of the LVLM architectures used in our experiments. These LVLMs respectively leverage the pre-trained vision encoder of the models we listed, which are all based on the Vision Transformer (ViT) (Dosovitskiy et al., 2020) architecture.

Table 8: Details of the LVLM architectures that we used in our paper.

| Model | Vision encoder | LLM |
|---|---|---|
| LLaVA (Liu et al., 2023d) | CLIP-L (Radford et al., 2021) | LLaMA-2-7B-Chat (Touvron et al., 2023) |
| LLaVA-v1.5 (Liu et al., 2023c) | CLIP-L-336px (Radford et al., 2021) | Vicuna-v1.5-7B (Chiang et al., 2023) |
| MiniGPT-v2 (Chen et al., 2023) | EVA-G (Fang et al., 2023) | LLaMA-2-7B-Chat (Touvron et al., 2023) |
| mPLUG-OWL2 (Ye et al., 2023) | CLIP-L (Radford et al., 2021) | LLaMA-2-7B (Touvron et al., 2023) |
| InstructBLIP (Dai et al., 2023a) | BLIP-2 (Li et al., 2023a) | Vicuna-v1.1-7B (Chiang et al., 2023) |

### C.2  DESCRIPTIONS ABOUT ADDITIONAL METRICS

In Figure 2, we evaluate the text quality of the outputs generated with MARINE using general metrics as follows:

- *BLEU* (Papineni et al., 2002) measures how well the generated translation matches the reference translations in terms of n-gram overlap.
- *ROUGH-L* (Lin, 2004) measures the quality of a machine-generated summary by comparing it to one or more reference summaries.
- *CIDEr* (Vedantam et al., 2015) assesses the quality of image captioning models. It focuses on evaluating how well the generated captions align with human consensus.
- *SPICE* (Anderson et al., 2016) focuses on assessing the semantic similarity between the generated captions and reference captions.

### C.3  PROMPT TEMPLATES

For each query, we randomly select a prompt template from the available template list, as shown in Table 9.

### C.4  DETAILS OF BASELINES

Specifically, the hyperparameters for LURE (Zhou et al., 2023), VCD (Leng et al., 2023), OPERA (Huang et al., 2023a) are reported in Table 10, 11 and 12 respectively. We strictly

Table 9: Details of the LVLM architectures that we used in our paper.

| Template Type | Prompt Template |
|---|---|
| MARINE-intersec | This image contains `<OBJECT_GROUNDING>`. Based on this, `<QUERY>` |
| | The image contains the following objects: `<OBJECT_GROUNDING>`. Given these detected objects, `<QUERY>` |
| | This image shows the following objects: `<OBJECT_GROUNDING>`. Using this information, `<QUERY>` |
| | The objects found in this image are: `<OBJECT_GROUNDING>`. Considering this list of objects, `<QUERY>` |
| POPE task | This image contains only the following objects: `<OBJECT_GROUNDING>`. Do not assume anything beyond these objects. Based solely on this list, `<QUERY>` |
| | The detected objects in the image are: `<OBJECT_GROUNDING>`. Answer based only on these objects. `<QUERY>` |
| | This image shows the following objects: `<OBJECT_GROUNDING>`. You must answer using only the objects in this list. Given these detected objects, `<QUERY>` |
| | The objects found in this image are limited to: `<OBJECT_GROUNDING>`. You should rely strictly on this list of objects and make no other guesses. Based on this, `<QUERY>` |
| MARINE-union | List of detected objects in the image: |
| | `<OBJECT_GROUNDING_A>` |
| | `<OBJECT_GROUNDING_B>` |
| | Based on the detected objects above, `<QUERY>` |
| | The most prominent objects detected are: |
| | `<OBJECT_GROUNDING_A>` |
| | `<OBJECT_GROUNDING_B>` |
| | Given these findings, `<QUERY>` |
| | The following objects were detected in the image: |
| | `<OBJECT_GROUNDING_A>` |
| | `<OBJECT_GROUNDING_B>` |
| | With this information, `<QUERY>` |
| | Here is a list of all objects detected in the image: |
| | `<OBJECT_GROUNDING_A>` |
| | `<OBJECT_GROUNDING_B>` |
| | Do not infer or hallucinate any additional objects. Using only the detected objects, `<QUERY>` |

followed the original implementations and default hyperparameters described in their papers to reproduce the results for each baseline.

Table 10: LURE (Zhou et al., 2023) Hyperparameter Settings

| Parameters | Value |
|---|---|
| Uncertainty Threshold $\gamma$ | 0.9 |
| Position Threshold $\iota$ | 0.8 |

Table 11: VCD (Leng et al., 2023) Hyperparameter Settings

| Parameters | Value |
|---|---|
| Amplification Factor $\alpha$ | 1 |
| Adaptive Plausibility Threshold | 0.1 |
| Diffusion Noise Step | 500 |

Table 12: OPERA (Huang et al., 2023a) Hyperparameter Settings

| Parameters | Value |
|---|---|
| Self-attention Weights Scale Factor $\theta$ | 50 |
| Attending Retrospection Threshold | 25 |
| Beam Size | 5 |
| Attention Candidates | 1 |
| Penalty Weights | 1 |

## C.5 EXPERIMENT SETTING FOR HALLUCINATION EVALUATIONS

Key factors that potentially affect the hallucination evaluation outcomes, including the evaluation dataset and prompt template, LVLM's sampling strategy and batched generation techniques, and guidance strength, are detailed in this section. The hyper-parameters setting for MARINE and overall experiment settings are shown in Table 13 and 14.

Table 13: `MARINE` Hyperparameter Settings. The settings are fixed depending on the question-answering tasks.

| Parameters | Value |
|---|---|
| *Guidance* | |
| Guidance Strength | 0.7 |
| Noise Intensity for DETR | 0.95 |
| Detect Threshold for RAM++ | 0.68 |
| *Generation* | |
| Max Token Length | 64 |
| Sampling | Greedy |
| Random Seed | 242 |

Table 14: Batch size for LVLM generation is fixed across all experiments unless otherwise noted. To expedite the evaluation process, we employed the batched generation. We avoid the negative impact of batched generation by adopting left padding if the LVLM does not explicitly assign the padding strategy for inference.

| Model | LLaVA | LLaVA-v1.5 | MiniGPTv | mPLUG-Owl2 | InstructBLIP |
|---|---|---|---|---|---|
| Batch Size | 16 | 4 | 32 | 16 | 16 |

**Experiment setting for CHAIR evaluation.** We adopt the same prompt "Generate a short caption of the image." as utilized by Li et al. (2023b). The hyperparameters are fixed, including a guidance strength of 0.7, noise intensity for DETR at 0.95, a detection threshold for RAM++ of 0.68, a maximum token length of 64, and a greedy sampling approach with a random seed of 242.
For the calculation of CHAIR metrics, we referenced the 80 object categories annotated in the MSCOCO dataset, following Rohrbach et al. (2018). Besides, we employed the synonym list from Lu et al. (2018) to align synonymous words in the generated text with MSCOCO object categories. Additionally, due to the cost considerations associated with the GPT-3.5 API, we limited our analysis to 200 samples for Woodpecker correction for each model and reported the result in Table 1.

**Experiment setting for POPE evaluation.** POPE is a flexible approach to evaluating hallucinations in LVLMs, which formulates a binary classification task by prompting LVLMs with questions such as "Is there a keyboard in this image?" to answer "yes" or "no". Following Li et al. (2023b), we created 3000 POPE questions across three datasets—500 images each from MSCOCO, A-OKVQA, and GQA for the POPE evaluation. We reported the adversarial settings in Table 2, the most challenging setting, which constructs POPE questions from the top-k most frequently co-occurring but absent objects. Additionally, in Table 4, we reported the average scores under random, popular, adversarial settings across MSCOCO, A-OKVQA, and GQA datasets. The full POPE results are in Tabel 15.
Similarly, we constrained our analysis to 200 samples for Woodpecker correction for each model due to the high costs associated with the GPT API. The outcomes of this analysis are detailed in Table 2.

**Experiment setting for GPT-4V-aided evaluation.** The GPT-4V-aided evaluation compares the outputs of two LVLM assistants using GPT-4V as a judge. We prompted GPT-4V to assess the quality of the generated outputs, scoring them out of 10 in two aspects:
- *Accuracy*: how accurately each assistant describes the image;
- *Detailedness*: the richness of necessary details in the response.

As shown in Figure 5, the assessment prompt template we used is slightly different from that of Yin et al. (2023). Specifically, we include the original question for a task-orientated evaluation and exclude prompts that describe Woodpecker-specific output formats like object bounding boxes. Examples of the GPT-4V-aid evaluation responses are illustrated in Figure 6 and 7. Besides, a fixed guidance strength of 0.5 was used in the evaluations in Table 3. Utilizing the `gpt-4-1106-vision-preview`, all final experiments were conducted between 01/01/2024-01/30/2024. As OpenAI continues to update its API, accessing different versions may result in slightly different values.

**Experiment setting for ablation study.** To explore different methods of integrating image-grounding models, we investigate the intersection and union of detected objects, with integration based on synonyms using the NLTK package.
To quantitatively assess the influence of guidance strength, we varied it from 0 to 1, as shown in Figure 10. These quantitative experiments were conducted using the same setting as those in CHAIR

evaluation. For qualitative analysis, exemplified in Figure 13 and 10, we selected guidance strength from a recommended range of $\gamma \in (0.3, 0.7)$.

---

**Prompt**

You are required to score the performance of two AI assistants in describing a given image. You should pay extra attention to the hallucination, which refers to the part of descriptions that are inconsistent with the image content, such as claiming the existence of something not present in the image.

Please rate the responses of the assistants on a scale of 1 to 10, where a higher score indicates better performance, according to the following criteria:

1: Accuracy: whether the response is accurate with respect to the image content. Responses with fewer hallucinations should be given higher scores.

2: Detailedness: whether the response is rich in necessary details. Note that hallucinated descriptions should not count as necessary details.

Please output a single line for each criterion, containing only two values indicating the scores for Assistant 1 and 2, respectively. The two scores are separated by a space. Following the scores, please provide an explanation of your evaluation, avoiding any potential bias and ensuring that the order in which the responses were presented does not affect your judgment.",

Please score the performance of two AI assistants in describing a given image following the given question.

Question:
{question}

Assistant 1:
{answer 1}

Assistant 2:
{answer 2}

Output format:
Accuracy:
Scores of the two answers:
Reason:

Detailedness:
Scores of the two answers:
Reason:

---

Figure 5: Prompt template for GPT-4V-aided evaluation. {question} is the original instruction; {answer 1} is the original response, and {answer 2} is the response generated by the LVLM using `MARINE-DETR` with a guidance strength of $0.5$.

## C.6 EXPERIMENT SETTING ON OTHER VISION-LANGUAGE TASKS

**Experiment setting for text quality analysis.** For text quality analysis, we adopted 90 visual questions from the LLaVA-QA90 task (including conversations, visual perceptions, and complex reasoning subtasks), and randomly selected 50 MSCOCO images for image captioning task. Following Liu et al. (2023d), we adpoted the response generated by text-only GPT-4 (0314) with the context captions/boxes provided. answers given by GPT-4 as references for LLaVA-QA90 task and used image captions provided in MSCOCO annotations as references for image captioning task.

In Table 16 and Table 17, we present a detailed evaluation on the image captioning task for both MSCOCO and LLaVA-QA90 using metrics including BLEU, ROUGE, CIDEr and SPICE. The corresponding figure result is shown in Figure 2.

**Experiment setting for latency analysis.** We compared our method with existing baselines in terms of the trade-off between inference cost and the effectiveness of reducing object hallucinations, as shown in Table 5. For post-correction baselines such as Woodpecker and LURE, we first prompted LLaVA (`llava-llama-2-7b-chat-lightning-preview`) to generate captions and then measure the latency of generating the corrected outputs. The total latency for post-correction baselines includes both the generation and correction processes. For decoding methods such as VCD, OPERA and our method, we measured the latency of LLaVA generating captions directly.

We prompted the models with "Generate a short caption of the image." on 500 MSCOCO images with a batch size of 1 and a maximum token length of 64, without any stopping criteria, using a single A6000 GPU. Then latency was calculated as the ratio of the number of output tokens and encoding and generation time.

**Prompt**
Question:
Can you elaborate on the elements of the picture provided?

Assistant 1 (LLaVA-v1.5):
The image features a desk with various items on it. There is a bookshelf with a few books, including a prominent one titled \"Administrative Law \" A blue and white tie is placed on the desk, adding a touch of color and formality to the scene. A pair of sunglasses is also resting on the desk, possibly belonging to someone who was recently there.

In addition to the books and tie, there is a cup and an apple on the desk, indicating that the person might have been enjoying a snack or drink while working or studying. A teddy bear is also present in the scene sitting on the desk, and adding a playful touch to the setting.

Assistant 2 (LLaVA-v1.5 w/ MARINE):
The image features a desk with various items on it. There is a book, possibly a law book, positioned on the left side of the desk. A blue and white tie is placed next to the book, adding a touch of color to the scene. On the right side of the desk, there is an apple, possibly a Granny Smith apple, and a pair of sunglasses.

Additionally, there is a stuffed teddy bear located at the top right corner of the desk, giving the scene a cozy and personal touch. A baseball glove is also present in the image, suggesting that the owner might have an interest in sports.

**GPT-4V Evaluation**
Accuracy:
Scores of the two answers: 4 7
Reason: Both assistants mention items that are present in the image, such as the book with the title "Administrative Law," the tie, the apple, the sunglasses, and the teddy bear. Assistant 1 hallucinates the presence of a cup, which is not visible in the image. Assistant 2 inaccurately describes the presence of a baseball glove, which is not in the image; the item mistaken for a glove is actually part of the teddy bear's outfit. Assistant 2 also avoids mentioning the specific title of the book, even though it is visible in the image.

Detailedness:
Scores of the two answers: 5 6
Reason: Assistant 1 provides a more detailed description of the scene by elaborating on the potential reasons for the items' presence, such as the person enjoying a snack or the teddy bear adding a playful touch, However, the mention of a cup, which is not in the image, detracts from the quality of the response. Assistant 2 provides a sufficient amount of detail without over-elaborating and makes a specific guess about the type of apple. However, the incorrect mention of a baseball glove and the lack of details about the book titles compared to Assistant 1 result in a slightly lower score.

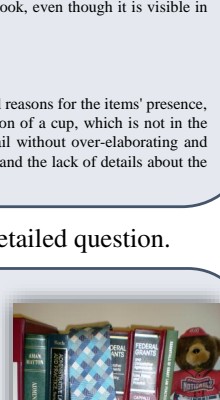

Figure 6: An example of GPT-4V-aided evaluation on a detailed question.

**Prompt**
Question:
What might be the purpose of this table arrangement?

Assistant 1 (LLaVA-v1.5):
The purpose of this table arrangement might be to create a comfortable and productive workspace. The table features a pair of sunglasses, a book, and an apple, which are essential items for reading, relaxation and snacking. The presence of the book and sunglasses suggests that the person might be taking a break from reading or working on a project, while the apple provides a healthy snack option. The combination of these items on the table indicates that the person values both work and relaxation, aiming to create a balance between the two.

Assistant 2 (LLaVA-v1.5 w/ MARINE):
The purpose of this table arrangement appears to be a combination of work and leisure. The presence of books. a sunglasses. a tie. and an apple suggests that the person using this space may be engaged in work or study, as indicated by the books. Additionally, the apple and sunglasses indicate that the person might be taking a break, enjoying a snack, or preparing for a more relaxed activity. The presence of the teddy bear adds a touch of playfulness and personalization to the setting, creating a cozy and inviting atmosphere.

**GPT-4V Evaluation**
Accuracy:
Scores of the two answers: 6 7
Reason: Both Assistants made a few errors in their descriptions, Assistant 1 incorrectly states that there\'s a book on the table, which is ambiguous; technically, the books are not on the table but more likely on a shelf next to it and the tie isn't mentioned, which is a significant element in the image. Assistant 2 correctly recognizes the presence of books, a tie, sunglasses, and an apple, along with the teddy bear dressed in a sports jersey. However Assistant 2 also refers to books in a slightly ambiguous way, implying they are on the table, and incorrectly describes the sunglasses as "a sunglasses" which is grammatically incorrect and could slightly mislead the context.
Detailedness:
Scores of the two answers: 6 7
Reason: Assistant 1 provides an explanation of the potential purpose of the table arrangement but misses out mentioning the tie and teddy bear both of which add context to the scenario depicted. The description lacks detail on the actual arrangement and how it specifically suggests a mixture of work and leisure. Assistant 2 includes more elements in the description, such as the tie and teddy bear, adding to the detail and mentioning relaxation and personalization, which gives a slightly better insight into the purpose of the table arrangement, but their response could also be more nuanced by acknowledging the role of each item.

Figure 7: An example of GPT-4V-aided evaluation on a complex question.

Table 15: Detailed POPE (Li et al., 2023b) results on three datasets (MSCOCO (Lin et al., 2014), A-OKVQA (Schwenk et al., 2022), GQA (Hudson & Manning, 2019)).

| Dataset | Type | Model | w/MARINE | Accuracy ↑ | Precision ↑ | Recall ↑ | F1 ↑ | Yes(%) |
|---|---|---|---|---|---|---|---|---|
| MSCOCO | Adversarial | LLaVA | ✗ | 51.8 | 50.9 | 99.5 | 67.4 | 97.7 |
| | | | ✓ | 66.9 | 61.7 | 89.1 | 72.9 | 72.3 |
| | | mPLUG-Owl2 | ✗ | 72.5 | 65.5 | 94.9 | 77.5 | 72.4 |
| | | | ✓ | 82.8 | 83.4 | 82.0 | 82.7 | 49.2 |
| | Popular | LLaVA | ✗ | 52.4 | 51.2 | 99.8 | 67.7 | 97.4 |
| | | | ✓ | 71.3 | 65.8 | 88.9 | 75.6 | 67.5 |
| | | mPLUG-Owl2 | ✗ | 75.8 | 68.7 | 94.9 | 79.7 | 69.0 |
| | | | ✓ | 85.6 | 88.4 | 82.0 | 85.1 | 46.4 |
| | Random | LLaVA | ✗ | 58.3 | 54.5 | 99.7 | 70.5 | 91.4 |
| | | | ✓ | 78.5 | 73.4 | 89.3 | 80.6 | 60.8 |
| | | mPLUG-Owl2 | ✗ | 81.8 | 75.2 | 94.9 | 83.9 | 63.1 |
| | | | ✓ | 88.1 | 93.4 | 81.9 | 87.3 | 43.9 |
| A-OKVQA | Adversial | LLaVA | ✗ | 50.0 | 50.0 | 99.5 | 66.6 | 99.5 |
| | | | ✓ | 56.3 | 53.6 | 94.3 | 68.3 | 88.1 |
| | | mPLUG-Owl2 | ✗ | 62.5 | 57.3 | 98.1 | 72.3 | 85.6 |
| | | | ✓ | 74.4 | 68.8 | 89.3 | 77.7 | 64.9 |
| | Popular | LLaVA | ✗ | 50.1 | 50.1 | 99.8 | 66.7 | 99.7 |
| | | | ✓ | 63.0 | 58.0 | 94.5 | 71.9 | 81.6 |
| | | mPLUG-Owl2 | ✗ | 69.1 | 62.1 | 97.9 | 76.0 | 78.9 |
| | | | ✓ | 82.5 | 78.8 | 89.1 | 83.6 | 56.5 |
| | Random | LLaVA | ✗ | 55.4 | 52.8 | 99.8 | 69.1 | 94.4 |
| | | | ✓ | 73.7 | 66.7 | 94.7 | 78.3 | 71.0 |
| | | mPLUG-Owl2 | ✗ | 77.2 | 69.2 | 98.2 | 81.2 | 71.0 |
| | | | ✓ | 89.2 | 89.2 | 89.3 | 89.2 | 50.1 |
| GQA | Adversial | LLaVA | ✗ | 50.3 | 50.1 | 99.8 | 66.8 | 99.5 |
| | | | ✓ | 54.4 | 52.5 | 93.8 | 67.3 | 89.4 |
| | | mPLUG-Owl2 | ✗ | 68.4 | 63.0 | 98.2 | 75.6 | 79.8 |
| | | | ✓ | 76.0 | 73.6 | 81.2 | 77.2 | 55.2 |
| | Popular | LLaVA | ✗ | 50.1 | 50.0 | 99.8 | 66.7 | 99.7 |
| | | | ✓ | 58.7 | 55.1 | 94.3 | 69.5 | 85.5 |
| | | mPLUG-Owl2 | ✗ | 70.6 | 63.8 | 94.9 | 76.3 | 74.4 |
| | | | ✓ | 77.6 | 75.6 | 81.3 | 78.4 | 53.8 |
| | Random | LLaVA | ✗ | 55.7 | 53.0 | 99.8 | 69.2 | 94.1 |
| | | | ✓ | 74.3 | 67.3 | 94.8 | 78.7 | 70.5 |
| | | mPLUG-Owl2 | ✗ | 82.0 | 75.2 | 95.5 | 84.1 | 63.5 |
| | | | ✓ | 86.8 | 91.5 | 81.3 | 86.1 | 44.4 |

Table 16: Performance on general metrics for the image captioning task, including BLEU (Papineni et al., 2002), ROUGE-L (Lin, 2004), CIDEr (Vedantam et al., 2015) and SPICE (Anderson et al., 2016) scores(%).

| Model | w/MARINE | BLEU_1 (↑) | BLEU_2 (↑) | BLEU_3 (↑) | BLEU_4 (↑) | ROUGE_L (↑) | CIDEr (↑) | SPICE (↑) |
|---|---|---|---|---|---|---|---|---|
| LLaVA | ✗ | 14.06 | 7.12 | 3.72 | 1.90 | 22.06 | 0.08 | 16.77 |
| | ✓ | 18.59 | 9.96 | 5.47 | 3.04 | 26.02 | 0.21 | 20.58 |
| mPLUG-Owl2 | ✗ | 39.91 | 25.16 | 16.57 | 11.24 | 36.26 | 1.05 | 26.82 |
| | ✓ | 39.51 | 24.37 | 15.93 | 10.70 | 36.01 | 1.03 | 27.42 |

Table 17: Performance on general metrics for the LLaVA-QA90 task, including BLEU (Papineni et al., 2002), ROUGE-L (Lin, 2004), CIDEr (Vedantam et al., 2015) and SPICE (Anderson et al., 2016) scores(%).

| Model | w/MARINE | BLEU_1 (↑) | BLEU_2 (↑) | BLEU_3 (↑) | BLEU_4 (↑) | ROUGE_L (↑) | CIDEr (↑) | SPICE (↑) |
|---|---|---|---|---|---|---|---|---|
| LLaVA | ✗ | 21.02 | 12.91 | 8.79 | 6.41 | 32.30 | 0.93 | 31.36 |
| | ✓ | 23.37 | 14.39 | 9.59 | 6.83 | 33.81 | 0.99 | 31.91 |
| mPLUG-Owl2 | ✗ | 44.50 | 28.57 | 19.58 | 14.43 | 40.24 | 1.46 | 40.51 |
| | ✓ | 45.82 | 28.87 | 19.24 | 13.70 | 38.54 | 1.29 | 38.70 |

# D ADDITIONAL EXPERIMENTS

## D.1 ADDITIONAL BASELINES

### D.1.1 DIRECT PROMPTING

We conducted additional experiments to compare our approach with a baseline that uses carefully engineered prompts designed to reduce hallucination:

*Describe the visible contents of this image in as much detail as possible without adding any information not clearly visible. Only mention objects, colors, shapes, and textures that can be directly*

Table 18: Comparison against carefully engineered prompts.

| Method | LLaVA | | | LLaVA-v1.5 | | | mPLUG-Owl2 | | |
|---|---|---|---|---|---|---|---|---|---|
| CHAIR | $C_s \downarrow$ | $C_i \downarrow$ | Recall $\uparrow$ | $C_s \downarrow$ | $C_i \downarrow$ | Recall $\uparrow$ | $C_s \downarrow$ | $C_i \downarrow$ | Recall $\uparrow$ |
| Original | 26.6 | 10.5 | 47.4 | 8.8 | 4.6 | 41.1 | 5.0 | 3.2 | 33.2 |
| Direct Prompting | 27.2 | 11.0 | 46.4 | 19.6 | 8.3 | **52.3** | 9.0 | 5.1 | **42.0** |
| Prompts as Additional Guidance | 37.4 | 10.5 | 50.4 | 12.6 | 5.9 | 44.6 | 6.6 | 3.9 | 40.4 |
| **MARINE (ours)** | **17.8** | **7.2** | **50.8** | **6.2** | **3.0** | 44.3 | **4.2** | **2.3** | 41.4 |

Table 19: Comparison against ensembling multiple LVLMs. We consider ensembling all possible combinations of the five LVLMs and report the average performance with the standard deviation.

| Method | Accuracy $\uparrow$ | F1 $\uparrow$ | Yes Ratio |
|---|---|---|---|
| Voting on 2 LVLMs | $76.7 \pm 7.8$ | $79.8 \pm 4.8$ | $62.6 \pm 11.2$ |
| **MARINE(ours)** | $79.9 \pm 7.4$ | $80.4 \pm 4.6$ | $51.1 \pm 12.6$ |
| Voting on 3 LVLMs | $83.2 \pm 1.2$ | $83.0 \pm 1.0$ | $48.6 \pm 2.8$ |

*observed in the image, avoiding assumptions about materials, functions, or contexts. If there are any uncertainties about what an object is, describe its visual characteristics (e.g., 'a circular object with a smooth surface') without inferring its purpose or identity. Avoid creative or hypothetical descriptions, and focus on observable details only.*

With two different settings:

- **Direct Prompting**: The original input query was replaced with the prompts as described.
- **Prompts as Additional Guidance**: We incorporated the prompt as supplemental context to guide the models in generating outputs.

The results demonstrate that incorporating the prompt can effectively enhance the recall performance for some models (e.g., LLaVA-v1.5 improves from 41.1 to 52.3 in recall). However, it did not consistently reduce hallucinations across all metrics and the performance on CHAIR scores (e.g., $C_s$, $C_i$) dropped. Meanwhile, MARINE significantly outperforms the prompting baseline approaches on CHAIR. We note the following differences between our method and prompting method:

- Prompting method depends heavily on the instruction-following ability of the model. While it might mitigate the hallucination to a mild extent for strong models (e.g., LLaVA-v1.5), it may cause a weak model to hallucinate even more (e.g., LLaVA). Models also require more sophisticated fine-tuning approaches to generate better and more precise response conditioned on the prompts, as discussed in Deng et al. (2024). In contrast, our method directly addresses deficiencies in the model's vision capabilities by introducing stronger vision guidance. This makes our approach more effective even for weaker models and more cost-efficient.
- Unlike prompting methods, which need to be tailored to specific tasks or datasets, our method generalizes effectively across models and datasets, reducing hallucinations while maintaining competitive recall.

### D.1.2 MAJORITY VOTING AMONG LVLMS

Ensembling different LVLMs is a strong baseline, as majority voting among models enhances robustness. However, it is less frequently used in practice due to the significant cost of acquiring and loading multiple LVLMs, which is substantially higher than that of additional vision models. In Table 19, we show the POPE results of the average performances of ensembling all possible combinations of 2 and 3 LVLMs among the 5 LVLMs that we experimented with. Notably, our method outperforms the ensemble of two LVLMs. While the ensemble of three LVLMs achieves higher scores, it comes with significantly higher computational and memory costs, making it much less practical for many real-world scenarios. Meanwhile, our method requires only 30% more GPU memory than the plain LVLM during inference. For large-batch inference and online chatbot deployment, ensembling multiple LVLMs is much less feasible due to the substantial increase in memory consumption. In contrast, MARINE remains an accessible, efficient, and effective approach.

Furthermore, since most LVLMs use the same CLIP model as their vision component, ensembling multiple LVLMs primarily combines the language outputs, aiming for consistency across different LLMs—a strategy proven effective in the textual domain. This makes the ensemble of different LVLMs complementary to our method, which focuses on enhancing the vision component by incorporating multiple vision models.

Table 20: Comparing LVLM with specialized models on image captioning.

| Model | w/ MARINE | Bleu-1 ↑ | Bleu-2 ↑ | Bleu-3 ↑ | Bleu-4 ↑ | ROUGE$_L$ ↑ | CIDEr ↑ | Average ↑ |
|---|---|---|---|---|---|---|---|---|
| GIT-Large-COCO | - | 35.68 | 23.01 | 15.52 | 10.85 | 33.43 | 1.12 | 19.94 |
| LLaVA | ✗ | 14.06 | 7.12 | 3.72 | 1.90 | 22.06 | 0.08 | 8.16 |
| LLaVA | ✓ | 18.59 | 9.96 | 5.47 | 3.04 | 26.02 | 0.21 | 10.55 |
| mPLUG-Owl2 | ✗ | 39.91 | 25.16 | 16.57 | 11.24 | 36.26 | 1.05 | 21.70 |
| mPLUG-Owl2 | ✓ | 39.51 | 24.37 | 15.93 | 10.70 | 36.01 | 1.03 | 21.26 |

Table 21: Experiments on dynamic guidance strength based on confidence scores on CHAIR metrics.

| Method | LLaVA | | | mPLUG-Owl2 | | |
|---|---|---|---|---|---|---|
| CHAIR | $C_s$ ↓ | $C_i$ ↓ | Recall ↑ | $C_s$ ↓ | $C_i$ ↓ | Recall ↑ |
| Fix Guidance Strength | 17.8 | 7.2 | **50.8** | **4.2** | **2.3** | **41.4** |
| Dynamic Guidance Strength | **14.8** | **6.5** | 49.9 | 5.0 | 2.6 | 41.0 |

Table 22: Experiments on dynamic guidance strength based on confidence scores on POPE metrics.

| Method | LLaVA | | | mPLUG-Owl2 | | |
|---|---|---|---|---|---|---|
| POPE | Accuracy ↑ | F1 ↑ | Yes Ratio | Accuracy ↑ | F1 ↑ | Yes Ratio |
| Fix Guidance Strength | 66.9 | 72.9 | 72.3 | 82.8 | 82.7 | 49.2 |
| Dynamic Guidance Strength | **71.97** | **74.48** | **59.83** | **83.3** | **83.2** | **49.4** |

Lastly, while ensembling LVLM outputs for yes/no questions in POPE is straightforward, extending this approach to a diverse range of tasks—including instruction-following tasks—is significantly more challenging. Developing a feasible and effective method for ensembling LVLMs in such contexts remains an unexplored area for future research. In contrast, MARINE easily generalizes to different question formats and tasks, demonstrating its versatility and practicality.

### D.1.3 DISUCSSION ON SPECIALIZED MODELS FOR IMAGE CAPTIONING

We extended our captioning quality evaluation (Table 16) to include the Generative Image-to-text Transformer (GIT) (Wang et al., 2022), a highly specialized non-LLM-based model (∼0.8B) pre-trained on 4M images and fine-tuned on the MSCOCO dataset (GIT-Large-COCO), as a baseline. GIT notably outperforms many traditional image captioning methods across multiple benchmarks and, at the time, achieved state-of-the-art results with a significant margin. The results are demonstrated in Table 20.

We make the following key notes:

- While GIT-Large-COCO outperforms the older LLaVA model as expected given its specialized training, newer LVLMs like mPLUG-Owl2 achieve better performance (21.70 vs 19.94 average score) even compared to specialized captioning models. This demonstrates the rapid advancement of LVLMs (based on poweful LLM backbone Llama) in exceeding task-specific models.

- As we previously pointed out in the paper (line 414-416), image captioning evaluation serves to demonstrate that our method, while primarily targeting hallucination mitigation, maintains LVLMs' performance on broader tasks without significant trade-offs. For LLaVA model, our method even further improves LLaVA's performance across all metrics.

- For LLaVA, MARINE improves performance across all metrics. For mPLUG-Owl2 with MARINE, performance remains largely stable and continues to outperform the specialized model.

The competitive performance of LVLMs, combined with their flexibility across multiple tasks, supports our focus on enhancing these models rather than specialized ones.

### D.2 DYNAMIC GUIDANCE STRENGTH

We conducted additional experiments on dynamic guidance strength based on confidence scores, evaluating both CHAIR and POPE metrics, as shown in Table 21 and 22. Specifically:

- **Fix Guidance Strength** uses a fixed guidance strength of 0.7, selected to balance hallucination reduction and instructions adherence.

- **Dynamic Guidance Strength** adjusts the guidance strength dynamically by mapping the mean confidence score ($s$) of the image-grounding models to a range of (0.4, 0.8) using the

Table 23: Inference Latency (ms/token) Comparison. We report both the latency and the ratio to the latency of greedy decoding of the original LVLM model.

|  | Greedy | LURE | Woodpecker* | VCD | OPERA | Offline **MARINE** | Online **MARINE** |
|---|---|---|---|---|---|---|---|
| Training Cost | 0 | 10min on A100 80G | 0 | 0 | 0 | 0 | 0 |
| Inference Latency | 26.3 (×1.0) | 179.9 (×6.84) | 94.5 (×3.59)* | 53.4 (×2.03) | 185.1 (×7.0) | **52.2 (×1.98)** | **52.23 (×1.985)** |

*Woodpecker requires GPT API key access and the latency may depend on OPENAI API.

Table 24: Peak GPU Memory Usage during Inference (GB) of MARINE compared to greedy decoding and VCD.

| **Metric** | Greedy | VCD | MARINE (Ours) |
|---|---|---|---|
| Peak GPU Memory Usage | 23.53 | 20.73 (×0.88) | 30.78 (×1.30) |

formula

$$\gamma' = 0.4 + \frac{(0.8 - 0.4) \cdot (s - s_{\min})}{s_{\max} - s_{\min}}.$$

A higher confidence score indicates more reliable guidance and corresponds to a stronger guidance strength. The results show that dynamic guidance improves performance for the weaker model LLaVA, which is more susceptible to noisy guidance. For stronger models like mPLUG-Owl2, a fixed guidance strength sufficiently mitigates object hallucinations with a robust performance.

### D.3 COMPREHENSIVE LATENCY AND MEMORY ANALYSIS

Visual prompts in MARINE are pre-generated prior to inference and were not included in the total latency reported in the original Table 5, which reflects an offline setting. We further add the online setting that accounts for the time required to process images with different visual encoders at inference time. Table 23 shows the updated latency comparison, which includes the latency for generating visual prompts. These results demonstrate that processing images adds only a negligible overhead to the overall latency.

We further measured the peak GPU memory usage throughout the inference for 500 image captioning questions using LLaVA model with batch size =16, max generation length = 64 tokens. The results are presented in Table 24. The result shows that during inference, the GPU memory usage increases by approximately 30% instead of doubling. Although we introduced additional vision models, they are relatively small compared to the large LLM backbone, resulting in only a modest increase in memory usage. Lastly, we'd highlight that our method is training-free, and thus we do not require additional memory or computation use for training, which would be significantly more demanding due to the computation of gradients.

### D.4 FURTHER STUDY ON GUIDANCE STRENGTH

Figure 8 illustrates how varying guidance strength affects the quality of LLaVA's output text in both the LLaVA-QA90 task and the image captioning task (maximum generation length = 256). Our experiments demonstrate that a guidance strength of 1 does not yield the best image captioning performance (i.e., quality of the generated text). In LLaVA-QA90 task, a guidance strength $\gamma$ between 0.5 and 0.7 provides the most enhanced text quality. This aligns with a common concern in methods employing classifier-free guidance, where excessively strong guidance can divert the generation process.

Additionally, we conducted experiments comparing the overall generation quality of LLaVA using GPT-4V as a judge, scoring outputs on a scale of 10 for accuracy and detail, which further supports our findings. The comparison between models with and without balancing the original LVLM branch is summarized in Table 25.

Figure 9 demonstrates how overly strong guidance can reduce instruction adherence by inducing the model to include unnecessary details from the image.

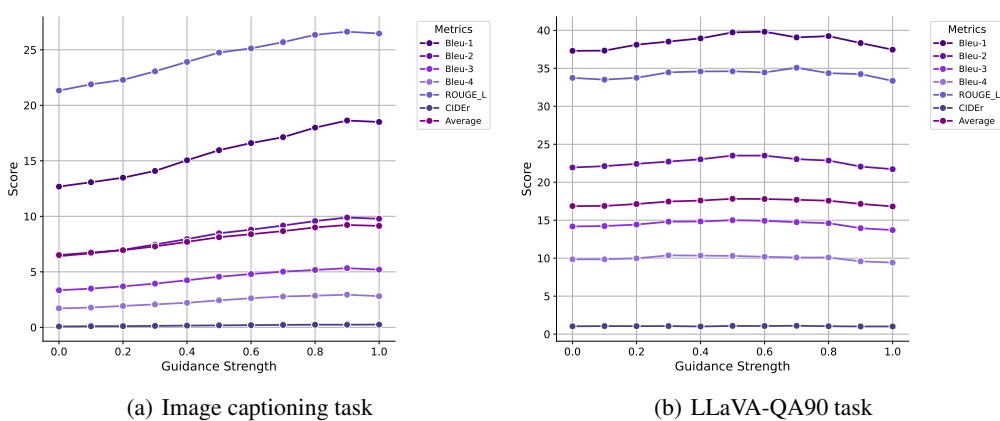

(a) Image captioning task        (b) LLaVA-QA90 task

Figure 8: The impact of guidance strength on the output text quality.

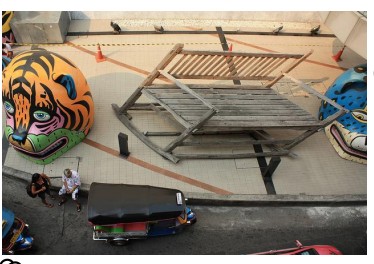

Question: How might the art installation encourage interaction among people?

**LLaVA w/ MARINE with balance guidance**

The art installation, which consists of a large, colorful, and intricately painted tiger sculpture, is likely to encourage interaction among people in several ways. Firstly, the vibrant colors and unique design of the tiger sculpture can attract attention and draw people towards it, making them curious about its origin and significance. This curiosity can lead to conversations and discussions among passersby, fostering social interactions and a sense of community.\n\n\nSecondly, the sculpture's size and prominence in the city's public space can encourage people to take a break from their daily routines and engage with the artwork. The presence of the tiger sculpture can create a sense of awe and wonder, encouraging people to pause and appreciate the beauty of the artwork. This can lead to a more relaxed and enjoyable atmosphere, as people take the time to admire the art and enjoy the surrounding environment.\n\n\nLastly, the sculpture's placement in the city's public space can also serve as a focal point for community events, such as art exhibitions, cultural festivals, or even street performances. The tiger sculpture can become a gather [reach the maximum generation length]

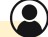 **GPT4 response as reference**

The art installation, featuring an oversized wooden bench and two massive animal art sculptures, encourages interaction among people by providing a visually intriguing and inviting space to sit or gather. People are naturally drawn to the unique and whimsical nature of these installations, sparking curiosity and conversation among those who encounter them. The oversized bench provides seating for multiple individuals, prompting people to sit down, rest, or socialize while admiring the giant painted feline heads and other artworks on display. Additionally, the installations capture the attention of passersby, creating a shared experience and a sense of community among those who visit the area. The artistic environment stimulates discussion and appreciation of the art, facilitating social interactions and connections among those who gather near the sculptures and oversized bench.

**LLaVA w/ MARINE with overly strong guidance**

The art installation, which features a large, colorful tiger sculpture and a car parked next to it, can encourage interaction among people in several ways. Firstly, the vibrant colors and unique design of the tiger sculpture can attract attention and draw people towards it, making them curious about its origin and purpose. This curiosity can lead to conversations and discussions among passersby, encouraging social interaction and engagement. Additionally, the presence of the car parked next to the sculpture can provide a focal point for people to gather and take photos, further fostering social interaction and creating a sense of community. The art installation can also serve as a backdrop for events or gatherings, such as art exhibitions, festivals, or even photo shoots, which can further encourage people to interact with each other and engage with the artwork. Overall, the art installation can serve as a catalyst for social interaction and community engagement, promoting a sense of connection and shared experience among people.

Figure 9: This case highlights that overly strong guidance can induce the model to prioritize providing exhaustive visual details from the image, even when such details are irrelevant to the specific instruction (e.g., "a car parked next to it"). In contrast, balanced guidance enables the model to maintain better adherence to the instruction while still utilizing the visual information effectively.

# E FURTHER MODEL PERFORMANCE ANALYSIS

## E.1 EFFECT OF MARINE ON LOGIT DISTRIBUTION.

In Figure 10, we illustrate a specific example that shows how MARINE influences the logit distribution of LVLMs during text generation. Specifically, MARINE is observed to selectively target the potential hallucinated tokens, reducing their original probabilities to mitigate the risk of hallucination in the generated text. For instance, in the provided example, the probability of "fork" is significantly lowered with MARINE, which would have originally resulted in a hallucinated object. Conversely, standard language elements such as "various", an adjective describing the overall image context, and

Table 25: Results of GPT-4V-aided evaluation. The accuracy and detailedness metrics are on a scale of 10, and a higher score indicates better performance. The symbols × and ✓ indicate performance metrics without and with our method, respectively.

| Task | Metrics | LLaVA | |
|---|---|---|---|
| | | ✗($\gamma = 1$) | ✓($\gamma = 0.7$) |
| LLaVA-QA90 | Acc ↑ | 5.52 | **5.79** |
| | Detail ↑ | 4.58 | **4.77** |
| Image Captioning | Acc ↑ | 6.06 | **6.22** |
| | Detail ↑ | 5.00 | **5.24** |

"with", a crucial preposition, maintain their original probabilities. This selective nature of modulation by `MARINE` ensures coherent and contextually relevant text generation that adheres to the instruction while effectively reducing hallucinations.

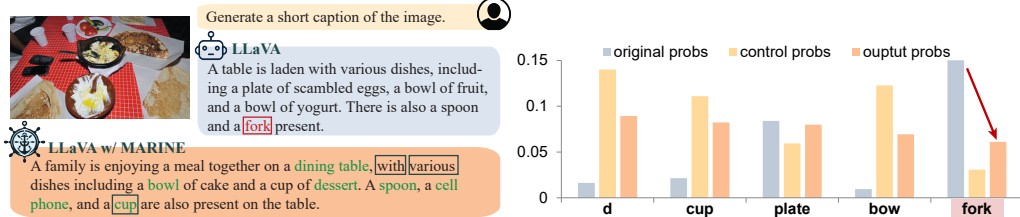

(a) An example of image description where the original LLaVA outputs a hallucinated object, "fork".

(b) The probability distributions at the token of the hallucinated word in the original, control, and `MARINE` outputs. `MARINE` effectively decreases the the probability of "fork".

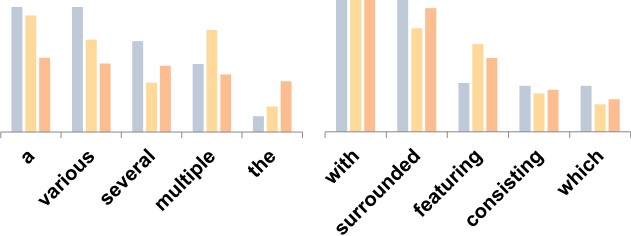

(c) Probabilities of non-hallucinated words remain the same, highlighting `MARINE`'s ability to preserve normal outputs.

Figure 10: This sample shows how `MARINE` controls logit distributions to mitigate hallucinations like "fork" while preserving the probabilities of "with", "various" during generation.

### E.2 DISCUSSION ON FINE-TUNING METHODS.

The examples depicted in Figure 11 illustrate that LURE, at times, fails to adhere to the given instructions when correcting LVLM generations. Despite receiving concise image descriptions generated based on instructions for short responses, LURE predominantly overwrites them with excessively long responses that contain information irrelevant to the instruction. Furthermore, LURE fails to adequately address the binary question format of POPE, as LURE fixates on extended descriptions without responding with "yes" or "no", making its evaluation using POPE impractical. This issue can be prevalent in small-scale fine-tuning methods, where the limited variety of the specifically tailored fine-tuning dataset harms the model's performance on other tasks. In contrast, the training-free approach of `MARINE` demonstrates effective mitigation of hallucinations across a variety of question formats.

### E.3 EXTENDED ANALYSIS IN ABLATION STUDY

Additional experimental results explore the noise intensity of object grounding features, which are examined across LLaVA, InstructBLIP, and mPLUG-Owl2, with findings presented in Figures 12, 14, and 15.

**Input**: Is there a book in the image?

**LLaVA**: **Yes**, there is a book in the image, and it is placed on the bed next to the doll.

**LURE**: There are two dolls in a bedroom scene, with one of them being the focus of attention. One of the dolls is sitting in a crib, while the other is lying in a baby carriage in the same room. A book can be seen on the bed behind the doll in the crib.

**Woodpecker**: **No**, there is no book in the image. The image shows two beds([0.294, 0.001, 0.999, 0.998];[0.0, 0.37, 0.658, 0.998]), and a doll([0.078, 0.439, 0.553, 0.92]) is placed on each bed. A teddy bear is next to the doll.

**MARINE**: **No**, there is no book in the image. The image features a doll in a crib, a doll on a bed, and a teddy bear.

Figure 11: Example responses to an image-question pair. The LURE-corrected output deviates from the original question, offering irrelevant descriptions without directly addressing the query. Woodpecker hallucinates the existence of two beds while there is only one bed in the figure. In contrast, MARINE maintains the original answer's style and adheres to the user's instruction while eliminating hallucination.

This variation is achieved by implementing four confidence thresholds (0.5, 0.7, 0.9, and 0.95) in the DETR model predictions (with MARINE-Truth serving as an ideal reference), where higher thresholds correspond to lesser, yet higher-quality, visual information. Our findings highlight two significant insights. Firstly, an increase in the quality of visual information correlates with a noticeable decrease in hallucinations produced by the LVLMs. A lower threshold, which allows for more visual information but also includes noisier content, could potentially result in an increased occurrence of hallucinations. Furthermore, lower-quality visual information is associated with enhanced Recall. This suggests that LVLMs under guidance, despite the presence of noisy visual inputs, tend to focus more on the visual details (i.e., objects), resulting in more elaborate descriptions.

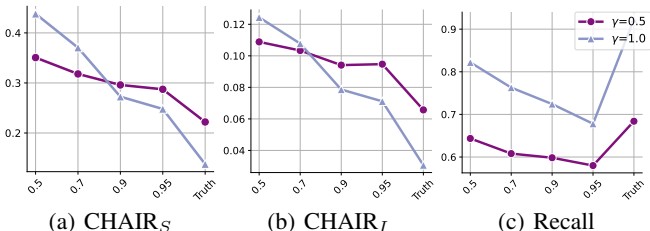

(a) CHAIR$_S$      (b) CHAIR$_I$      (c) Recall

Figure 12: LLaVA's performance on CHAIR according to different noise intensity of object grounding features in MARINE. We consider four confidence thresholds (0.5, 0.7, 0.9, and 0.95) for DETR to vary the noise intensity.

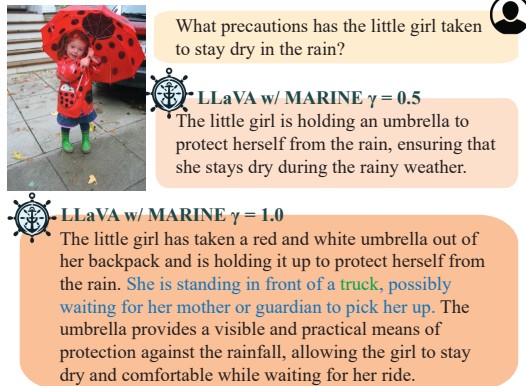

What precautions has the little girl taken to stay dry in the rain?

**LLaVA w/ MARINE γ = 0.5**
The little girl is holding an umbrella to protect herself from the rain, ensuring that she stays dry during the rainy weather.

**LLaVA w/ MARINE γ = 1.0**
The little girl has taken a red and white umbrella out of her backpack and is holding it up to protect herself from the rain. She is standing in front of a truck, possibly waiting for her mother or guardian to pick her up. The umbrella provides a visible and practical means of protection against the rainfall, allowing the girl to stay dry and comfortable while waiting for her ride.

Figure 13: An example of the negative impact of excessive guidance on LVLM's ability to follow instructions accurately. While the response with $\gamma = 1$ identifies more existing objects, it introduces irrelevant information to the instruction.

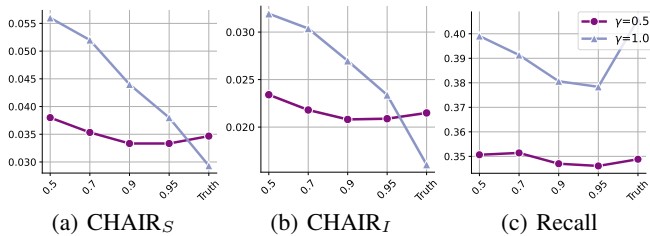

(a) CHAIR$_S$     (b) CHAIR$_I$     (c) Recall

Figure 14: InstructBLIP's performance on CHAIR according to different noise intensity of object grounding features in `MARINE`. We consider four confidence thresholds (0.5, 0.7, 0.9, and 0.95) for DETR to vary the noise intensity, with `MARINE`-Truth serving as an ideal reference.

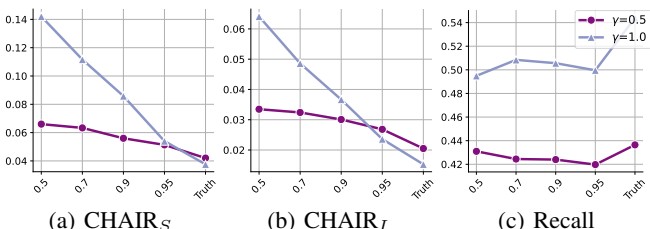

(a) CHAIR$_S$     (b) CHAIR$_I$     (c) Recall

Figure 15: mPLUG-Owl2's performance on CHAIR according to different noise intensity of object grounding features in `MARINE`. We consider four confidence thresholds (0.5, 0.7, 0.9, and 0.95) for DETR to vary the noise intensity, with `MARINE`-Truth serving as an ideal reference.

### E.4 MORE CASE STUDIES

In Figures 6 and 7, we present examples of GPT-4V-aided evaluations based on the outputs of LLaVA-v1.5 and LLaVA-v1.5 with `MARINE`. In Figures 16, 17, 18, 19, and 20, we present examples of the outputs from LURE (Zhou et al., 2023), Woodpecker (Yin et al., 2023) and `MARINE` on different tasks further validate our arguments in the paper.

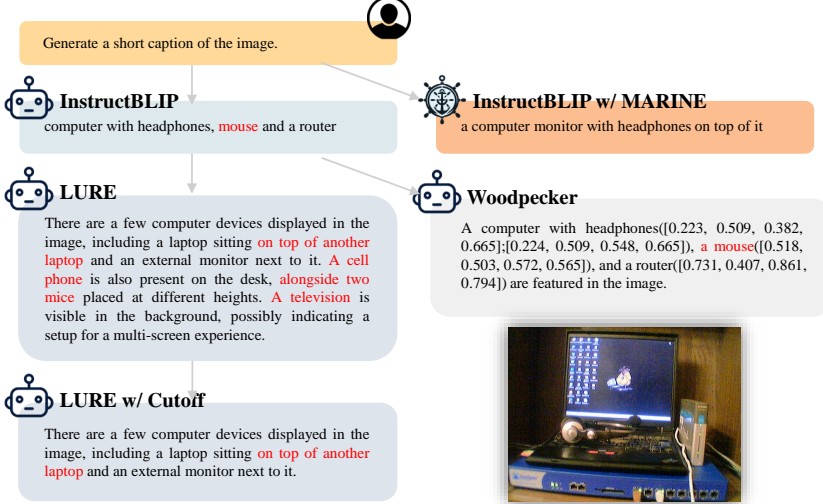

Figure 16: A comparison of responses from baseline models and our `MARINE` in an image description task. It illustrates `MARINE`'s superior ability to reduce hallucinations, in contrast to LURE and Woodpecker, which fail to effectively address hallucinations and sometimes even increase hallucinated content. This example highlights the strengths of our correct-during-generation framework over post-correction approaches, showcasing its efficiency, preservation of original style, and enhanced adherence to instructions.

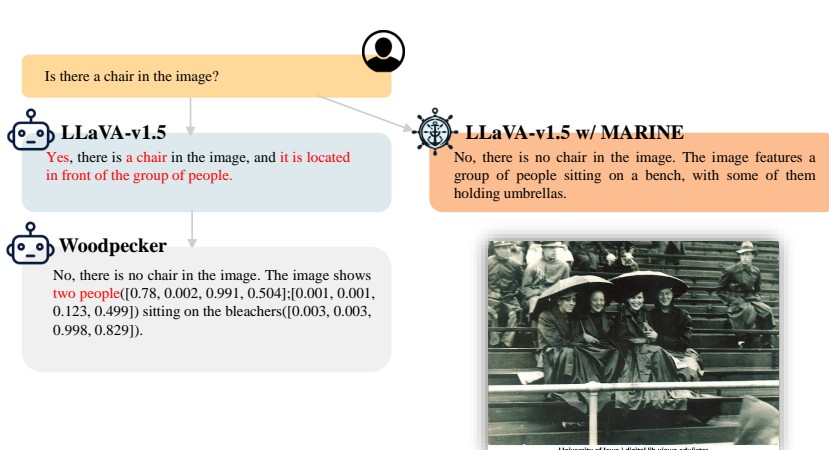

Figure 17: A comparison of responses from Woodpecker and our MARINE in POPE "yes-or-no" task.

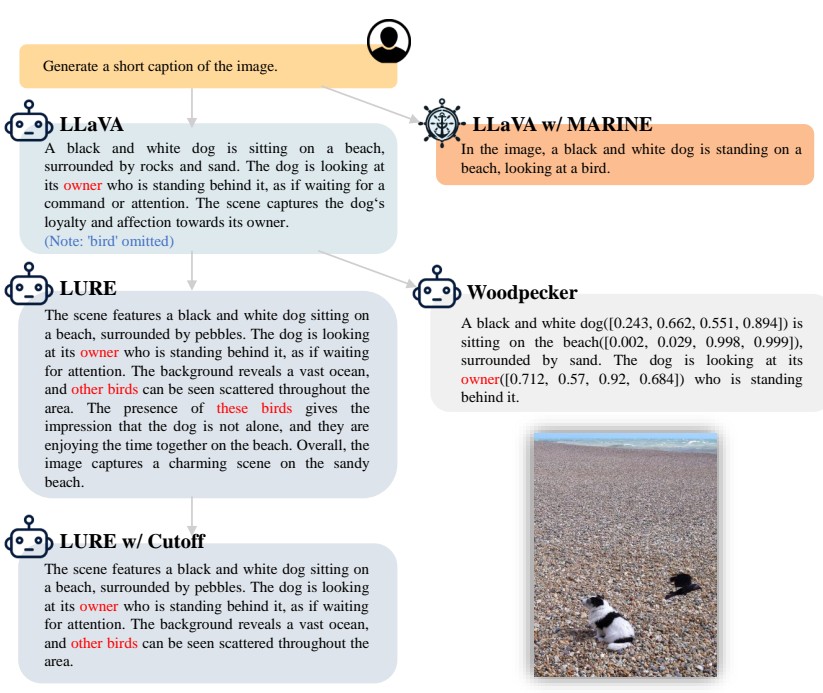

Figure 18: A comparison of responses from baseline models and our MARINE in an image description task. MARINE effectively reduces hallucinations and accurately includes the previously omitted object, 'bird', enhancing the description with essential details.

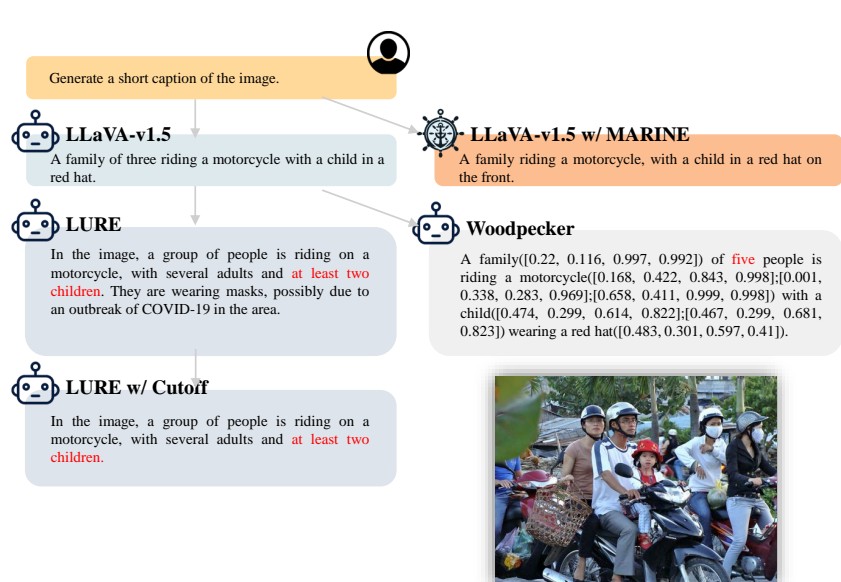

Figure 19: A comparison of responses from baseline models and our MARINE in an image description task.

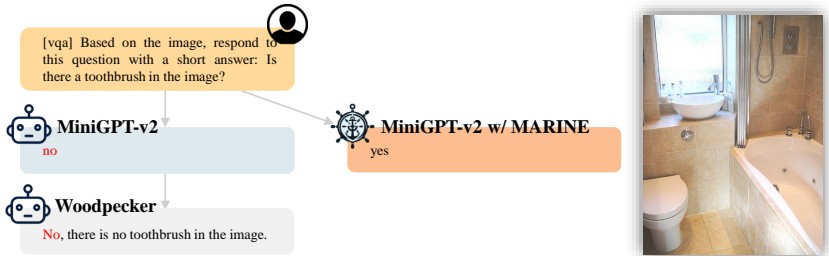

Figure 20: A comparison of responses from baseline models and our MARINE in POPE "yes-or-no" task. MiniGPT-v2 provides a concise response without referencing any objects. Under these circumstances, Woodpecker is unable to perform corrections via GPT-3.5 due to missing visual details. MARINE, however, successfully corrects the response while retaining MiniGPT-v2's style.

