# OpenReview forum: "Mitigating Object Hallucination in Large Vision-Language Models via Image-Grounded Guidance"
_ICLR.cc/2025/Conference — Submitted to ICLR 2025_

### Official Review · Reviewer_8B3T · 2024-10-27

**Soundness:** 3
**Presentation:** 3
**Contribution:** 3
**Rating:** 6
**Confidence:** 5

**Summary:**

The paper proposes a training-free method to eliminate hallucinations in Large Vision Language Models (LVLMs) by utilizing guidance prompts generated from grounding models, combined with original inputs, using classifier-free guidance to mitigate model hallucinations.

**Strengths:**

1. The authors provide code and comparisons for all evaluation outputs, which significantly enhances the work's credibility and technical merit.

2. The proposed method is straightforward, simple, and theoretically well-supported.

3. The experimental section is solid, demonstrating performance improvements across various benchmarks on different models.

**Weaknesses:**

At this current time, using CFG in VLMs, particularly for hallucination reduction, is not novel. The authors only discuss Visual Contrastive Decoding (VCD) without addressing or comparing with other relevant methods [1][2][3]. While understandable, as most of these methods use CFG with $\gamma>1$, this limitation should be addressed.

## References

1. "Paying more attention to image: A training-free method for alleviating hallucination in lvlms"
2. "Contrastive region guidance: Improving grounding in vision-language models without training"
3. "Prompt Highlighter: Interactive Control for Multi-Modal LLMs"

**Questions:**

Regarding the methodology:
As shown in Fig.3, theoretically, $\gamma=1$ already provides sufficient information in most cases. Why is it necessary to balance the original branch? The significance of CFG balance seems unclear. I recommend:
   - Additional experimental validation
   - Presentation of specific cases
   - Clarification of novelty, as if improvements can be achieved simply by adding extra prompt information, the paper's contribution may appear limited

---

> ### Author Response · Authors · 2024-11-21
> **Rebuttal to Reviewer 8B3T - Part 1**
>
> Thank you for your constructive feedback. Please find our detailed response below for the questions raised in the review.
>
> ---
>
> ### W1: Discussions on other relevant methods using CFG
>
> Thank you for mentioning these references; while they are very relevant, we note that they are concurrent works to ours. We have included them in the concurrent work discussion in our revised manuscript on lines 127-135 and a detailed discussion in Appendix B lines 825-829. Specifically:
> - Similarity: The methodologies of both MARINE and these concurrent works [1-3] can be formulated as approaches using Classifier-Free Guidance (CFG) aimed at improving the factuality of LVLMs.
> - Differences: These concurrent approaches differ in the focus of their method designs within the larger framework of CFG. Specifically:
>     - [1] addresses the issue of “Text Inertia”, where LVLMs produce the same hallucinated object descriptions regardless of whether an image is actually provided.
>     - [2] focuses on fine-grained guidance and especially operates at the sub-image level.
>     - [3] adjusts the attention weights of the visual tokens within the attention mechanism to control the focus of the generation on the highlighted parts.
>
> We highlight that MARINE operates at the image level and uniquely employs a vision toolbox to ensemble information from multiple vision models, producing guidance that reaches a consensus among the models for more accurate results.
>
> Lastly, we emphasize that these approaches are complementary to each other, and integrating them could pave the way for a more effective multimodal prompt-following strategy in future research.
>
> [1] Paying more attention to image: A training-free method for alleviating hallucination in lvlms
>
> [2] Contrastive region guidance: Improving grounding in vision-language models without training
>
> [3] Prompt Highlighter: Interactive Control for Multi-Modal LLMs

---

> ### Author Response · Authors · 2024-11-21
> **Rebuttal to Reviewer 8B3T - Part 2**
>
> ### Q1: Reasons to balance the original branch.
>
> Thank you for your valuable suggestion. While increasing the guidance strength leads to a notable decrease in the CHAIR score, we have observed that overly strong guidance can negatively affect the model's ability to follow instructions, as illustrated in Figure 12.
>
> ---
>
> **Q1.1: Additional experimental validation for CFG with different $\gamma$.**
>
> In response to your suggestion, we conducted additional experiments to further validate our findings. Figure 8 of the updated revision illustrates how varying guidance strength affects the quality of LLaVA's output text in both the LLaVA-QA90 task and the image captioning task (maximum generation length = 256), evaluated using metrics such as BLEU, ROUGE-L, and CIDEr. Due to space constraints, we summarize a subset of the results in the table below for the LLaVA-QA90 task.
>
> | $\gamma$ | Bleu-1 $\uparrow$ | Bleu-2 $\uparrow$ | Bleu-3 $\uparrow$ | Bleu-4 $\uparrow$ | ROUGE_L $\uparrow$ | CIDEr $\uparrow$ | average $\uparrow$  |
> |------|--------|--------|--------|--------|---------|-------|-------|
> | 0    | 37.3   | 21.9   | 14.2   | 9.9    | 33.7    | 1.03  | 16.9  |
> | 0.2  | 38.1   | 22.4   | 14.4   | 10.0   | 33.8    | 1.05  | 17.1  |
> | 0.4  | 38.9   | 23.0   | 14.8   | **10.3**   | **34.6**    | 1.01  | 17.6  |
> | 0.6  | **39.8**   | **23.5**   | **14.9**   | 10.2   | 34.5    | **1.07**  | **17.8**  |
> | 0.8  | 39.2   | 22.9   | 14.6   | 10.1   | 34.4    | 1.04  | 17.6  |
> | 1    | 37.4   | 21.7   | 13.7   | 9.4    | 33.4    | 1.01  | 16.8  |
>
> Our experiments demonstrate that a guidance strength of 1 does not yield the best image captioning performance (i.e., quality of the generated text). In LLaVA-QA90, a guidance strength $\gamma$ between 0.5 and 0.7 provides the most enhanced text quality. This aligns with a common concern in methods employing classifier-free guidance, where excessively strong guidance can divert the generation process.
>
> Additionally, we conducted experiments comparing the overall generation quality of LLaVA using GPT-4V as a judge, scoring outputs on a scale of 10 for accuracy and detail, which further supports our findings. The comparison between models with (✅) and without (❌) balancing the original LVLM branch is summarized below and Table 25 of our revision:
>
> | **Task** | **Metrics** | **❌ (γ = 1)** | **✅ (γ = 0.7)** |
> |------------|----------------|-----------|------------|
> | **LLaVA-QA90** | **Accuracy↑** | 5.52 | **5.79** |
> | | **Detail ↑** | 4.58 | **4.77** |
> | **Image Captioning** | **Accuracy↑** | 6.06 | **6.22** |
> | | **Detail ↑** | 5.00 | **5.24** |
>
> In summary, our additional experiments confirm that setting γ = 1 results in decreased text quality, both quantitatively across several evaluation metrics and qualitatively as assessed by GPT-4V evaluation.
>
> ---
>
> **Q1.2: Presentation of specific cases with and without balancing the original branch.**
>
> We analyzed specific cases to illustrate the effect of overly strong guidance.  Figure 9 of the updated revision demonstrates how overly strong guidance can reduce instruction adherence by inducing the model to include unnecessary details from the image.
> - **Question**: *How might the art installation encourage interaction among people?*
> - **LLaVA with overly strong guidance**: *"The art installation, which features a large, colorful tiger sculpture and a car parked next to it, can encourage interaction among people..."*
>
> This case highlights that overly strong guidance can induce the model to prioritize providing exhaustive visual details from the image, even when such details are irrelevant to the specific instruction (e.g., "*a car parked next to it*"). In contrast, balanced guidance enables the model to maintain better adherence to the instruction while still utilizing the visual information effectively.

---

> > ### Author Response · Authors · 2024-11-23
> > **Invitation for discussion**
> >
> > Dear Reviewer 8B3T
> >
> > Thank you again for your valuable feedback. We have carefully addressed your concerns and conducted additional experiments to provide further clarity on our methodology and findings. Specifically:
> >
> > - W1: We included a discussion on relevant concurrent works that falls under the general CFG framework in the revised manuscript, highlighting similarities and differences. We emphasized how MARINE complements these methods through unique ensembling techniques for robust vision guidance.
> > - Q1 & Q1.1: We conducted extensive experiments varying the guidance strength ($\gamma$), demonstrating that a balanced $\gamma$ improves both the quality and adherence of the generated text across tasks like LLaVA-QA90 and image captioning.
> > - Q1.2: We presented specific cases illustrating the drawbacks of excessive guidance, such as unnecessary detail inclusion, and showed how balanced guidance enables better instruction adherence while leveraging visual information effectively.
> >
> > We hope these additional experiments and clarifications address your concerns and highlight the robustness and practicality of our approach. Please let us know if there are further questions or additional details we can provide. Thank you again for your thoughtful feedback and consideration.

---

> > > ### Comment · Reviewer_8B3T · 2024-11-25
> > >
> > > Thank you for the author's active and detailed response. This rebuttal specifically addresses almost all of my questions. I choose to raise my score to 6 because I believe the paper has not yet found a good **general method** for enhancing VLM capabilities through additional guidance at this stage. Since the paper itself uses fixed general visual tools (i.e., grounding descriptions) as the guidance branch, this will inevitably lead to a reduction in instruction-following ability under the condition of CFG strength. This is also why I think the focus of the paper is not broad enough.

---

> > > > ### Author Response · Authors · 2024-11-25
> > > > **Thank you for your reply!**
> > > >
> > > > Thank you for raising your score and providing the thoughtful feedback! We are glad that our rebuttal addressed most of your concerns.
> > > >
> > > > Regarding the instruction-following ability, our experiments in Figure 8 demonstrated that proper guidance strength, as a hyperparameter, effectively reduces object hallucinations *almost without* compromising caption or VQA quality.
> > > >
> > > > While a broader investigation into general methods for LVLMs beyond object hallucination is indeed an important direction for future work, our paper specifically focuses on the well-defined but underexplored challenge of object hallucination and introduces a simple yet effective solution.
> > > >
> > > > Thank you again for acknowledging our rebuttal and increasing your recommendation accordingly.

---

### Official Review · Reviewer_JbpG · 2024-10-28

**Soundness:** 2
**Presentation:** 2
**Contribution:** 2
**Rating:** 5
**Confidence:** 4

**Summary:**

This paper presents a training-free and API-free framework named MARINE to reduce object hallucinations for LVLMs, with the help of open-source vision models to extract object-level information. Then MARINE integrates information from multiple vision models to refine the visual context provided to the LVLM in a contrastive decoding way. Comprehensive evaluations across multiple LVLMs and datasets demonstrate that MARINE reduces object hallucinations effectively.

**Strengths:**

1. The paper is well-written and easy to follow.
2. The core idea of this paper is simple yet effective.
3. MARINE demonstrates significant improvements over existing methods in reducing hallucinations across various metrics and multiple LVLMs.
4. MARINE’s design allows integration with different vision models, enhancing its flexibility and robustness in handling diverse contexts.

**Weaknesses:**

1. In Tables 1 and 2, introducing ground-truth objects (i.e., MARINE-Truth) leads to a significant performance drop in some metrics, which is the exact opposite of the core idea of the paper. Furthermore, the statement in Line 361 ‘The reference method, MARINE-Truth, generally achieves the strongest results.’ is not aligned with the reported results. Please provide a detailed explanation for this discrepancy and clarify why introducing ground-truth objects cause the performance drop.
2. MARINE’s performance depends on the quality of the external image-grounded models. Limitations in the accuracy of these models may impact MARINE’s effectiveness.
3. As Table 6 shows, using a single model leads to a significant performance drop, even worse than greedy decoding. This result is counter-intuitive and the authors should do a more in-depth analysis for this phenomenon. Good to provide additional experiments that could help explain this phenomenon as well.
4. This paper only discusses the additional inference latency of different method in Table 5. However, the additional GPU memory consumption should also be discussed. Please provide GPU memory usage data for each method in a format similar to Table 5,  which would give a more comprehensive view of the computational requirements.
5. While MARINE effectively mitigates object hallucination, its application is specifically focused on this issue within LVLMs. Extending it to address broader types of hallucinations in general LVLMs remains a challenge.

**Questions:**

Please refer to the weakness section.

---

> ### Author Response · Authors · 2024-11-21
> **Rebuttal to Reviewer JbpG - Part 1**
>
> Thank you for the detailed feedback. Please find our responses below. We hope that our clarifications and additional experiments resolve the concerns and any potential misunderstandings.
>
> ---
>
> ### Q1: Explore the reason why MARINE-Truth’s recall score are significantly lower than MARINE
> Thank you for highlighting the numerical errors in Tables 1 and 2; we have corrected them in the revised paper. Specifically, we updated the recall values for LLaVA and MiniGPT-4v in Table 1 and fixed the swapped F1 and Yes Ratio values in Table 2. With these corrections, MARINE-Truth generally achieves better performance, aligning with our intuition and the statement in Line 361.
>
> ---
>
> ### Q2: MARINE’s performance depends on the quality of the external image-grounded models.
> Regarding the question, we’d like to first emphasize that the performance of the current LVLMs significantly depends on the quality of the vision models integrated into their architectures. This dependency motivated us to propose MARINE, which introduces additional vision guidance to enhance the LVLM's ability to accurately recognize visual information. Our approach thus involves leveraging strong external guidance by ensembling information from multiple advanced vision models specialized in information extraction. The commonly used CLIP model in LVLM, on the other hand, focuses on vision-language alignment.
>
> The importance of the external vision model’s quality is indeed significant, as confirmed by our extended ablation studies (Figures 10, 12, and 13). By examining the impact of noise intensity in object grounding features, we demonstrate the critical role of precise object grounding information and illustrate how noise can degrade performance.
>
> This underscores:
> 1. **Why our method works**: These vision models are highly specialized and outperform CLIP in object detection, providing precise object grounding information that helps mitigate hallucinations.
> 2. **Why we adopt ensembling**: By aggregating visual information from multiple image-grounded models, we can reduce the impact of noisy or inaccurate guidance, thereby improving overall robustness.
>
> ---
>
> ### Q3: Using a single model leads to a significant performance drop, even worse than greedy decoding
>
> We would like to first point out that using a single model **effectively reduces hallucinations in most cases**. In our ablation study of Table 6, for example, MARINE-RAM successfully reduces $C_S$ and $C_I$ for LLaVA-v1.5, achieving the second-best performance while MARINE achieves the best. Although using a single model may slightly underperform compared to greedy decoding in some instances, this inconsistent improvement is also observed in many of the compared baselines for certain models and metrics. This motivates our approach of ensemble multiple vision models to reach a consensus in guidance, thereby achieving the best performance.
>
> This finding was aimed to highlight the following two key points:
> 1) **The importance of ensembling in MARINE**: Single models lack the robustness provided by ensembling, making them more susceptible to inaccurate or misleading guidance. On the other hand, ensembling provides a more precise object grounding in guidance.
> 2) **The limitation of the early LLaVA model**: The relatively weak robustness of this early version makes it more vulnerable to noisy guidance, leading to degraded performance.
>
> To gain a deeper understanding of this phenomenon, we conducted an extended ablation study on yes/no questions using POPE metrics. The results below include subscripts indicating performance gains compared to greedy decoding:
>
> | **Model** | **LLaVA-$Accuracy\uparrow$** | **LLaVA-$F1\uparrow$**| **LLaVA-Yes Ratio** | **mPLUG-Owl2-$Accuracy\uparrow$** | **mPLUG-Owl2-$F1\uparrow$**  | **mPLUG-Owl2-Yes Ratio**     |
> |----|------|----|----|----|----|-----|
> | **Ensembling Models** | | | | |  | |
> | **MARINE (ours)** | **$66.9_{+15.1}$** | **$72.9_{+5.5}$**   | **$72.3_{-25.4}$**  | **$82.8_{+10.3}$** | **$82.7_{+5.2}$**    | **$49.2_{-23.2}$**   |
> |--|
> | **Single Models**     |       |         |          |              |             |             |
> | MARINE-DETR only      | $73.1_{+21.3}$                 | $77.8_{+10.5}$               | $71.3_{-26.5}$             | $75.9_{+3.4}$                     | $80.1_{+2.6}$                 | $71.2_{-1.2}$                |
> | MARINE-RAM only       | $62.1_{+10.3}$                 | $71.8_{+4.4}$                | $84.4_{-13.3}$             | $78.7_{+6.2}$                     | $80.9_{+3.4}$                 | $61.8_{-10.7}$               |
>
> *Note: The ideal Yes Ratio for a non-biased LVLM is 50%.*
>
> These results demonstrate that while single models can provide valuable visual grounding information, our ensemble method is more robust and yields stronger overall improvements.

---

> > ### Author Response · Authors · 2024-11-21
> > **Rebuttal to Reviewer JbpG - Part 2**
> >
> > ### Q4: GPU memory usage comparison.
> >
> > In response to your suggestion, we measured the peak GPU memory usage throughout the inference for 500 image captioning questions using LLaVA model with batch size =16, max generation length = 64 tokens. The results are presented in the table below and in our updated revision (Table 24):
> >
> > | Metric    | Greedy  | VCD       | MARINE (Ours) |
> > |-------------|---------|--------------------|-----------------------|
> > | Peak GPU Memory Usage during Inference (GB) | 23.53   | 20.73  (x0.88)       | 30.78   ( x1.30)    |
> >
> > The result shows that during inference, the GPU memory usage increases by approximately 30% instead of doubling. Although we introduced additional vision models, they are relatively small compared to the large LLM backbone, resulting in only a modest increase in memory usage. Lastly, we’d highlight that our method is training-free, and thus we do not require additional memory or computation use for training, which would be significantly more demanding due to the computation of gradients.
> >
> > ---
> >
> > ### Q5: Extending it to address broader types of hallucinations in general LVLMs remains a challenge.
> >
> > Thank you for raising this question. The direction regarding extending our method to address other types of hallucination is indeed crucial. While we focused on object hallucination mitigation, our method essentially proposes ensembling information/context from other vision models to provide useful guidance to improve model truthfulness. It’s naturally applicable to use our method and ensemble different vision models for performance improvement on different tasks (such as to improve VQA performance).

---

> > > ### Author Response · Authors · 2024-11-23
> > > **Invitation for discussion**
> > >
> > > Dear Reviewer JbpG,
> > >
> > > Thank you for your detailed feedback and the opportunity to address your concerns further. We hope our responses and additional experiments have clarified key aspects of our work:
> > >
> > > - Q1: We corrected numerical errors in Tables 1 and 2, confirming that MARINE-Truth aligns with expectations and typically outperforms MARINE, as clarified in Line 361.
> > > - Q2: We emphasized the critical role of external vision models in MARINE and presented ablation studies (Figures 10, 12, and 13) to show the impact of noise in object grounding features. These results validate MARINE’s robustness in leveraging ensemble guidance to enhance performance.
> > > - Q3: Through extended ablations, we demonstrated that while single models can reduce hallucinations, they lack the robustness provided by ensembling, which helps MARINE achieve stronger performance across tasks. We also clarified the limitations of early LVLMs like LLaVA in handling noisy guidance.
> > > - Q4: We provided detailed GPU memory usage comparisons, showing a modest increase (30%) during inference for MARINE. Our approach remains computationally efficient, as it is training-free and avoids additional memory demands for training.
> > > - Q5: We highlighted MARINE’s potential to address broader hallucination types by adapting its ensembling framework for other tasks like VQA, showing the method’s extensibility and versatility.
> > >
> > > We hope these clarifications address your concerns and demonstrate the robustness and practicality of our approach. If you have additional questions or need further details, we would be happy to provide them. Thank you again for your valuable feedback and consideration.

---

> > > > ### Comment · Reviewer_JbpG · 2024-11-26
> > > >
> > > > Thank you for your thoughtful rebuttal. I appreciate the effort you have put into addressing my comments and concerns. After carefully reviewing your responses, I acknowledge the points raised and the improvements made to improve your work.
> > > >
> > > > However, after further consideration, I believe my initial evaluation and rating remain consistent with my overall assessment of the work. While your rebuttal provides helpful context, it does not fully address the key aspects of my concerns. Thank you once again for your efforts.

---

> > > > > ### Author Response · Authors · 2024-11-26
> > > > > **Inquiry for further context**
> > > > >
> > > > > Thank you for your acknowledgment of the improvements made in response to your initial comments, as well as your recognition that our rebuttal provides helpful context.
> > > > >
> > > > > However, we are unsure which specific key aspects of your concerns you feel remain unaddressed. We would greatly appreciate further clarification of the unresolved points.

---

> > > > > > ### Author Response · Authors · 2024-11-29
> > > > > >
> > > > > > Dear reviewer JbpG,
> > > > > >
> > > > > > Thank you again for taking the time to review our paper.
> > > > > >
> > > > > > In our earlier message, we sought clarification on the specific aspects of your concerns that you feel were not fully addressed in our rebuttal. Unfortunately, we have not received a response, and we want to ensure we fully understand and resolve any outstanding issues.
> > > > > >
> > > > > > Could you provide further context on the key concerns? We aim to leverage this opportunity to make meaningful improvements and avoid any potential misunderstandings.

---

### Official Review · Reviewer_HeC9 · 2024-10-29

**Soundness:** 4
**Presentation:** 3
**Contribution:** 3
**Rating:** 8
**Confidence:** 4

**Summary:**

This paper tackles the object hallucination problem in LVLMs by utilizing 2D detectors (DETR, RAM++) to provide visual prompts. It proposes that these grounding models provide visual features with higher quality, and combining multiple prompt sources yields better performance.

**Strengths:**

1. The method description is very clear and easy to understand.
2. Bringing 2D visual feature seems to be a good way to prompt LVLMs to mitigate their inner issues.
3. The results seem to be good.

**Weaknesses:**

1. While the paper is in very good shape, there are minor errors in writing: in section 4.2, ‘or a rule-based algorithms like XXX’, I think ‘XXX’ might be a typo. Please correct me on this if the authors would like to refer to some work that does not have a reference yet.
2. Applying 2D features from pre-trained models to prompt downstream tasks (e.g., 3D detection, 2D few-shot detection) is not a completely novel idea. It would be wonderful to see more explanation on this method’s difference from direct prompting.

**Questions:**

1. Are the pre-trained DETR and RAM++ models being pre-trained before adding to the pipeline? Additionally, does the training vocabulary of DETR and RAM++ influence the performance? Also, I am curious whether the visual prompts are pre-generated prior to the LLM inference or if this 2D prompt-generation time is included in the total inference time.

2. I am also curious about how the performance differs from when using CLIP-extracted features as input prompts compared to DETR and RAM++. Also, there are other 2D feature extractors, including SAM, which has the potential to provide more fine-grained features.

3. In Table 1, some of the MARINE-Truth’s recall score are significantly lower than MARINE. i.e. the recall score in LLaVA. It will be interesting to see why this is the case since the the access to ground truth guidance is guaranteed.

4. This manuscript mentions two issues in section 4: 1) deficiencies in the visual encoder providing insufficient visual information. 2) misalignment between visual and textual domains. From my current understanding, this work addresses the first issue by providing 2D visual features. It will be nice to see more explanation on whether this method addresses the second issue.

---

> ### Author Response · Authors · 2024-11-21
> **Rebuttal to Reviewer HeC9 - Part 1**
>
> We're grateful for your strong support and helpful suggestions on our work, for which we have accordingly made modifications to our manuscript. Please find our detailed response below for the questions raised in the review.
>
> ---
>
> ### W1 Minor errors in writing.
>
> Thank you for pointing out these minor errors in our writing. We have corrected them in line 236 of the revised version of the manuscript.
>
> ---
>
> ### W2: Discussions on the difference with direct prompting methods
>
> Thank you for the suggestion. We conducted additional experiments to quantitatively compare with direct prompting and present the results in the table below:
> | Method  | LLAVA-$C_s\downarrow$ | LLAVA-$C_i\downarrow$ | LLAVA-$Recall\uparrow$ | LLaVA-v1.5-$C_s\downarrow$ | LLaVA-v1.5-$C_i\downarrow$ | LLaVA-v1.5-$Recall\uparrow$ | mPLUG-Owl2-$C_s\downarrow$ | mPLUG-Owl2-$C_i\downarrow$ | mPLUG-Owl2-$Recall\uparrow$ |
> |---------|---------|---------|----------|--------|--------|----------|---------|--------|----------|
> | Original  | 26.6        | 10.5        | 47.4         | 8.8              | 4.6              | 41.1             | 6.2                       | 3.4                       | 38.8                       |
> | Direct Prompting          | 27.2        | 11.0        | 46.4         | 19.6             | 8.3              | **52.3**             | 9.0              | 5.1              | **42.0**             |
> | Prompts as Additional Guidance | 37.4        | 10.5        | 50.4         | 12.6             | 5.9              | 44.6             | 6.6              | 3.9              | 40.4             |
> | **MARINE** (ours)  | **17.8**    | **7.2**     | **50.8**     | **6.2**          | **3.0**          | 44.3         | **4.2**          | **2.3**          | 41.4         |
>
> Specifically, we evaluated:
> - **Direct Prompting**: The original input query was replaced with the following prompt: *Describe the visible contents of this image in as much detail as possible without adding any information not clearly visible. Only mention objects, colors, shapes, and textures that can be directly observed in the image, avoiding assumptions about materials, functions, or contexts. If there are any uncertainties about what an object is, describe its visual characteristics (e.g., 'a circular object with a smooth surface') without inferring its purpose or identity. Avoid creative or hypothetical descriptions, and focus on observable details only.*
> - **Prompts as Additional Guidance**: We incorporated the prompt as supplemental context to guide the models in generating outputs.
> The results demonstrate that incorporating the prompt can effectively enhance the recall performance for some models (e.g., LLaVA-v1.5 improves from 41.1 to 52.3 in recall). However, it did not consistently reduce hallucinations across all metrics and the performance on CHAIR scores (e.g., $C_s$, $C_i$) dropped. Meanwhile, MARINE significantly outperforms the prompting baseline approaches on CHAIR.
>
> We note the following differences between our method and prompting method:
> - Prompting method depends heavily on the instruction-following ability of the model. While it might mitigate the hallucination to a mild extent for strong models (e.g., LLaVA-v1.5), it may cause a weak model to hallucinate even more (e.g., LLaVA). Models also require more sophisticated fine-tuning approaches to generate better and more precise responses conditioned on the prompts, as discussed in [1]. In contrast, our method directly addresses deficiencies in the model’s vision capabilities by introducing stronger vision guidance. This makes our approach more effective even for weaker models and more cost-efficient.
> - Unlike prompting methods, which need to be tailored to specific tasks or datasets, our method generalizes effectively across models and datasets, reducing hallucinations while maintaining competitive recall.
>
> [1] Enhancing Large Vision Language Models with Self-Training on Image Comprehension

---

> > ### Author Response · Authors · 2024-11-21
> > **Rebuttal to Reviewer HeC9 - Part 2**
> >
> > ### Q1.1. Are the pre-trained DETR and RAM++ models being pre-trained before adding to the pipeline?
> >
> > We directly adopted the pre-trained DETR and RAM++ models without any additional fine-tuning on our task.
> >
> > ---
> >
> > ### Q1.2. Does the training vocabulary of DETR and RAM++ influence the performance?
> >
> > Thank you for bringing up this discussion. DETR [1] is pretrained on predefined MSCOCO object vocabulary, while RAM++ [2] is a state-of-the-art open-set image tagging model. As we ensemble the guidance from both models, the resulting vocabulary will be within the DETR’s predefined training vocabulary. Meanwhile, the MARINE-RAM variant (Table 6) and MARINE-union variant (Table 7) demonstrate the fully open-vocabulary approaches. While these methods reduce hallucinations, the best performance is still achieved with the ensemble that includes DETR.
> >
> > For closed-vocabulary detectors like DETR, we emphasize that their closed vocabulary set does not constrain the LVLM’s free-form responses. Instead, these detectors help reduce hallucination by providing guidance within the set that sufficiently encompasses frequent objects commonly observed in natural images. Thus in practice, we observe effective mitigation for datasets other than MSCOCO, as supported by the significant improvement observed in POPE results on the A-OKVQA [2] shown in Table 15.
> >
> > [1] Open-Set Image Tagging with Multi-Grained Text Supervision
> >
> > [2] A-OKVQA: A Benchmark for Visual Question Answering using World Knowledge
> >
> > ---
> >
> > ### Q2. Are visual prompts pre-generated prior to the LLM inference or included during inference time?
> > Thank you for raising this point. Visual prompts in MARINE are pre-generated prior to inference and were not included in the total latency reported in the original Table 5, which reflects an offline setting. We further add the online setting that accounts for the time required to process images with different visual encoders at inference time.
> > Below is the updated latency comparison table, which includes the latency for generating visual prompts:
> >
> > |                      | Greedy        | LURE              | Woodpecker*             | VCD               | OPERA            | **Offline MARINE**   | **Online MARINE**  |
> > |-------------------|---------------|-------------------|-------------------------|-------------------|------------------|---------------------|---------------------|
> > | **Training Cost**              | 0             | 10min on A100 80G | 0                       | 0                 | 0                | 0                   | 0                   |
> > | **Inference Latency (ms/token)** | 26.3 (×1.0) | 179.9 (×6.84) | 94.5 (×3.59)* | 53.4 (×2.03) | 185.1 (×7.0) | **52.2 (×1.98)** | **52.23 (×1.985)** |
> >
> >
> > *Woodpecker requires GPT API key access, and the latency may depend on OPENAI API.
> >
> > These results demonstrate that processing images adds only a negligible overhead to the overall latency. We have updated the paper to include these additional results (Table 23) per your suggestion.
> >
> > ---
> >
> > ### Q3.1: How does the performance differ from when using CLIP-extracted features as input prompts compared to DETR and RAM++?
> >
> > The LVLMs we considered (e.g., LLaVA models) rely solely on CLIP-extracted features as visual prompts. Replacing DETR and RAM++ with CLIP-extracted features in MARINE results in a decoding process focused on self-consistency. While our approach emphasizes stronger guidance to address CLIP’s intrinsic deficiencies, this variant prioritizes reducing potential misalignments within the alignment layer by explicitly extracting the objects from CLIP. This is indeed a compelling direction that could integrate seamlessly into our framework for further exploration.
> >
> > ---
> >
> > ### Q3.2: Other extractors like SAM have the potential to provide more fine-grained features.
> >
> > We fully agree. Our vision toolbox is designed to enable the easy integration of advanced feature extractors, such as SAM, to enhance guidance quality. Exploring the inclusion of such extractors represents an exciting and promising avenue for future work.

---

> > > ### Author Response · Authors · 2024-11-21
> > > **Rebuttal to Reviewer HeC9 - Part 3**
> > >
> > > ### Q4: Exploring the reason why MARINE-Truth’s recall score are significantly lower than MARINE
> > >
> > > Thank you for highlighting the numerical errors in Tables 1 and 2; we have corrected them in the revised paper. Specifically, we updated the recall values for LLaVA and MiniGPT-4v in Table 1 and fixed the swapped F1 and Yes Ratio values in Table 2. With these corrections, MARINE-Truth generally achieves better performance, aligning with our intuition and the statement in Line 361.
> > >
> > > ---
> > >
> > > ### Q5. This work primarily addresses the first issue by providing 2D visual features. It will be nice to see more explanation on whether this method addresses the second issue.
> > >
> > > Thanks for the insightful comments. As we discussed in Q3.1, our current approach of incorporating additional vision models for ensemble indeed places more emphasis on addressing deficiencies in the visual encoder and provides richer and more precise visual information. However, MARINE's framework also incorporates elements that target the second issue of alignment.
> > >
> > > Specifically, the component in MARINE’s framework of directly extracting explicit object-level information and using it for guidance helps address the second issue. This "direct alignment" approach avoids potential losses that may occur during the linear transformation of the trainable alignment layer in LVLMs, by maintaining and emphasizing object-level information intactly during the vision-language mapping. This enables a variant of MARINE that directly and solely targets the alignment issue, as you also suggested in Q3.1, to use the explicit objects found with CLIP features. Incorporating CLIP into the ensemble toolbox would therefore further improve the mitigation at alignment aspect.

---

> > > > ### Comment · Reviewer_HeC9 · 2024-11-22
> > > > **Thank the authors for the response**
> > > >
> > > > Dear Authors,
> > > >
> > > > I have carefully read your responses and appreciate your modifications, explanation and experiments. Your rebuttal addressed my concerns. After reading other reviewers’ comments, I agree that there are some limitations. However, I would like to keep my original rating for its method’s simplicity and performance.

---

> > > > > ### Author Response · Authors · 2024-11-22
> > > > > **Thank you for your feedback!**
> > > > >
> > > > > We appreciate the reviewers for recognizing our contribution!

---

### Official Review · Reviewer_YMx7 · 2024-11-01

**Soundness:** 3
**Presentation:** 3
**Contribution:** 2
**Rating:** 5
**Confidence:** 4

**Summary:**

This paper proposes MARINE, a training-free and API-free method to address object hallucinations in LVLMs. MARINE integrates an object detector into the pipeline for reliable object identification. The detector's outputs are used as a guidance (a form of controllable generation). The balance between the original LVLM's text generation and the detector's guidance is controlled by a parameter gamma. Compared with baselines, the proposed method shows performance gain on LLM hallucination metrics (CHAIR, POPE).

**Strengths:**

1. The proposed method combines the complimentary strengths of perception models (object detectors) and generation models. Solving object hallucinations with an object detector is a convincing direction.
2. The proposed method is clean and self-contained. It requires no training and external API calls.
3. Presentation of the paper, including experiments and evaluation is well put together. Comparisons of object detectors and LVLMs, sources of guidances, inference latency analysis make the paper informative and comprehensive.

**Weaknesses:**

The major concerns are about novelty:

1. The proposed method essentially combines object detection outputs with LVLM generation using a weighted scheme on the token logits. Novelty and technical contribution can be enhanced if more sophisticated methods of bridging the detector and the LLM is proposed. Some rough examples include (1) Does gamma have to be a constant? Any algorithms to dynamically tune gamma based on factors like confidence scores? (2) The detector produces a fixed set of classes. This closed vocabulary nature is different from the LLM. Any thoughts on bridging the vocabulary gap?

2. Using perception models like object detectors to help image-to-text generation were studied in earlier non-LLM methods. Once LVLMs are adapted by the proposed approach, the authors didn't discuss whether the results outperform these specialized models in image captioning and VQA (Table 16). A comparison to related specialized models can better contextualize the results.

**Questions:**

1. I'm wondering if the proposed method is applied on SOTA models fine-tuned on MSCOCO (not generic VLLMs), will the performance improve. Do the authors have any thoughts on this?

---

> ### Author Response · Authors · 2024-11-21
> **Rebuttal to Reviewer YMx7 - Part 1**
>
> Thank you for the constructive feedback. Please find our responses below. We hope that our clarifications and additional experiments resolve the concerns.
>
> ### W1: Further discussion on contributions.
> Thank you for your constructive suggestions. In response to the specified questions, we **a)** added experiments introducing a component that dynamically adjusts $\gamma$ based on confidence scores and **b)** clarified that the inclusion of RAM++ in our method does provide a flexible open-set vocabulary. We begin with clarification of our contributions. In the following W1a and W1b, we provide detailed results and explanations.
>
> **Clarification of our contributions**
>
> The novelty of MARINE lies in its training-free and API-free guidance framework with an ensemble mechanism that integrates any open-source vision models. This offers an efficient alternative to training LVLMs with different vision backbones from scratch, saving resources while enabling the ensemble of specialized vision models without excessive memory and computation costs. Recent works [1-4] have also explored training-free approaches to guide LVLMs, highlighting the importance of effective training-free methods. In lines 825-829 of our revision, we also provided detailed discussion on the concurrent training-free methods.
>
> [1] Mitigating Object Hallucinations in Large Vision-Language Models through Visual Contrastive Decoding
>
> [2] Paying more attention to image: A training-free method for alleviating hallucination in lvlms
>
> [3] Contrastive region guidance: Improving grounding in vision-language models without training
>
> [4] Prompt Highlighter: Interactive Control for Multi-Modal LLMs
>
> ---
>
> **W1a: Dynamically tuning the guidance strength ($\gamma$)**
>
> Thank you for your insightful suggestion. In response, we conducted additional experiments on dynamic guidance strength based on confidence scores, evaluating both CHAIR and POPE metrics, as shown in the tables below and included as Table 21 and 22 in appendix D.2 of our updated revision.
>
> | Method    | LLAVA-$C_s\downarrow$ | LLAVA-$C_i\downarrow$ | LLAVA-$Recall\uparrow$ | mPLUG-Owl2-$C_s\downarrow$ | mPLUG-Owl2-$C_i\downarrow$ | mPLUG-Owl2-$Recall\uparrow$ |
> |------------|-------------|-------------|--------------|-------------|------------|-------------------|
> | Fix Guidance Strength        | 17.8        | 7.2         | **50.8**         | **4.2**              | **2.3**              | **41.4**             |
> | **Dynamic Guidance Strength**  | **14.8**    | **6.5**     | 49.9     | 5.0          | 2.6          | 41.0         |
>
> | Method               | LLAVA-$Accuracy\uparrow$ | LLAVA-$F1\uparrow$ | LLAVA-Yes Ratio | mPLUG-Owl2-$Accuracy\uparrow$ | mPLUG-Owl2-$F1\uparrow$ | mPLUG-Owl2-Yes Ratio |
> |-------------|-----------|----------|-----------|----------------|---------------|----------------|
> | Fix Guidance Strength         | 66.9      | 72.9     | 72.3      | 82.8     | 82.7         | 49.2     |
> | **Dynamic Guidance Strength**  | **71.97** | **74.48**| **59.83** | **83.3**       | **83.2**      | **49.4**       |
>
> Specifically:
> - **Fix Guidance Strength** uses a fixed guidance strength of 0.7, selected to balance hallucination reduction and instructions adherence.
> - **Dynamic Guidance Strength** adjusts the guidance strength dynamically by mapping the mean confidence score ($s$) of the image-grounding models to a range of (0.4, 0.8) using the formula $$\gamma' = 0.4 + \frac{(0.8 - 0.4)\cdot (s - s_{\text{min}})}{s_{\text{max}} - s_{\text{min}}}.$$
> A higher confidence score indicates more reliable guidance and corresponds to a stronger guidance strength.
>
> The results show that dynamic guidance improves performance for the weaker model LLaVA, which is more susceptible to noisy guidance. For stronger models like mPLUG-Owl2, a fixed guidance strength sufficiently mitigates object hallucinations with a robust performance. Lastly, we consider setting the guidance strength as a tunable parameter a promising direction for future work.

---

> ### Author Response · Authors · 2024-11-21
> **Rebuttal to Reviewer YMx7 - Part 2**
>
> **W1b: This closed vocabulary nature of detectors is different from the LLM.**
>
> Thank you for highlighting this point. For clarification, we indeed aimed to address the detector’s fixed set of vocabularies by incorporating RAM++ [1], a state-of-the-art open-set image tagging model to effectively mitigate object hallucinations in more general scenarios. The MARINE-RAM variant (Table 6) and MARINE-union variant (Table 7) demonstrate the effectiveness of open-vocabulary approaches. While these methods reduce hallucinations, the best performance is still achieved with the ensemble that includes DETR. However, MARINE’s flexibility allows for seamless integration of advances in open-vocabulary detection models and holds promise for further improvement through ensembling multiple open-vocabulary models.
>
> For closed-vocabulary detectors, we emphasize that their closed vocabulary set does not constrain the LVLM’s free-form responses. Instead, these detectors help reduce hallucination during LVLM decoding by providing more grounded guidance within the predefined set. The set, moreover, sufficiently encompasses frequent objects commonly observed in natural images in many datasets other than MSCOCO. This generalizability is supported by the significant improvement observed in POPE results on the A-OKVQA [2] shown in Table 15, which demonstrates the ability of our method to generalize beyond MSCOCO.
>
> [1] Open-Set Image Tagging with Multi-Grained Text Supervision
>
> [2] A-OKVQA: A Benchmark for Visual Question Answering using World Knowledge
>
> ---
>
> ### W2: Comparing with related specialized non-LLM models in image captioning.
>
> Thank you for your suggestion. We extended our captioning quality evaluation (Table 16) to include the **Generative Image-to-text Transformer (GIT)** [1], a highly specialized non-LLM-based model (~0.8B) pretrained on 4M images and fine-tuned on the MSCOCO dataset (GIT-Large-COCO [2]), as a baseline. GIT notably outperforms many traditional image captioning methods across multiple benchmarks and, at the time, achieved state-of-the-art results with a significant margin.
>
> The results are as follows and also in Table 20 Appendix D.1.3 of our updated revision:
> | Model          | w/ Method   |   Bleu-1 $\uparrow$ | Bleu-2 $\uparrow$ | Bleu-3 $\uparrow$ | Bleu-4 $\uparrow$ | ROUGE_L $\uparrow$ | CIDEr $\uparrow$ | Average $\uparrow$  |
> |:---------------|:------------|---------:|---------:|---------:|---------:|----------:|--------:|----------:|
> | GIT-Large-COCO | ❌          |    35.68 |    23.01 |    15.52 |    10.85 |     33.43 |    1.12 |    19.94 |
> | LLaVA          | ❌          |    14.06 |     7.12 |     3.72 |     1.90 |     22.06 |    0.08 |     8.16 |
> | LLaVA          | ✅          |    18.59 |     9.96 |     5.47 |     3.04 |     26.02 |    0.21 |    10.55 |
> | mPLUG-Owl2     | ❌          |    39.91 |    25.16 |    16.57 |    11.24 |     36.26 |    1.05 |    21.70 |
> | mPLUG-Owl2     | ✅          |    39.51 |    24.37 |    15.93 |    10.70 |     36.01 |    1.03 |    21.26 |
>
> We make the following key notes:
> 1. While GIT-Large-COCO outperforms the older LLaVA model as expected given its specialized training, newer LVLMs like mPLUG-Owl2 achieve better performance (21.70 vs 19.94 average score) even compared to specialized captioning models. This demonstrates the rapid advancement of LVLMs (based on powerful LLM backbone Llama) in exceeding task-specific models.
> 2. As we previously pointed out in the paper (line 414-416), image captioning evaluation serves to demonstrate that our method, while primarily targeting hallucination mitigation, maintains LVLMs' performance on broader tasks without significant trade-offs. For the LLaVA model, our method even further improves LLaVA's performance across all metrics.
> 3. For LLaVA, MARINE improves performance across all metrics. For mPLUG-Owl2 with MARINE, performance remains largely stable and continues to outperform the specialized model.
>
> The competitive performance of LVLMs, combined with their flexibility across multiple tasks, supports our focus on enhancing these models rather than specialized ones. Lastly, we fully agree that including specialized methods could better contextualize the results, and have included them in Table 20 of our revision.
>
> [1] GIT: A Generative Image-to-text Transformer for Vision and Language
>
> [2] https://huggingface.co/microsoft/git-large-coco

---

> > ### Author Response · Authors · 2024-11-21
> > **Rebuttal to Reviewer YMx7 - Part 3**
> >
> > ### Q1: Could MARINE be applied to SOTA models fine-tuned on MSCOCO?
> >
> > Thank you for the question. We address it in the following three folds to clarify any potential confusion.
> >
> > **Q1.1 Clarification on the training data for current LVLMs and their common vision backbone CLIP.**
> >
> > We would like to clarify that **MSCOCO images are included in both the training of CLIP (the vision backbone) and the LVLMs we considered**. CLIP's pre-training involves a large corpus of image-caption pairs, including those from MSCOCO, to align the visual and textual embedding spaces using a contrastive loss. This alignment ensures that paired embeddings are close while unpaired ones are distant. More detailed examples of using MSCOCO for CLIP pre-training can be found [here](https://github.com/revantteotia/clip-training/blob/main/zero_shot_eval_output/coco_trained_clip_observations.md).
> >
> > Moreover, LVLMs that utilize the pre-trained CLIP vision backbone also include instruction-following data with images from MSCOCO. For example, the LLaVA models' 665k visual instruction tuning data contains the entire set of images from the MSCOCO train2017 split, paired with specific instruction-response pairs designed for these images. Details are available [here](https://github.com/haotian-liu/LLaVA?tab=readme-ov-file#visual-instruction-tuning). Therefore, MSCOCO images are in-distribution data for these LVLMs.
> >
> > Lastly, while both CLIP and object detection models like DETR are trained on MSCOCO, they are optimized for different objectives. This difference motivates our method to incorporate various vision model capabilities through guidance in a training-free manner.
> >
> > **Q1.2 Clarification on the purpose of LVLM training and why they are not designed to specialize in one image dataset.**
> >
> > LVLMs are designed to perform a wide range of tasks across diverse domains by leveraging extensive and varied training data. Their training aims to develop general understanding and instruction-following capabilities rather than specializing in specific datasets like MSCOCO. Specializing solely on MSCOCO could limit their abilities and reduce performance on user-case data. Therefore, LVLMs are typically trained on diverse datasets to maintain versatility and robustness in strong performance across different tasks and inputs.
> >
> > **Q1.3 Future research on applying MARINE to non-LLM methods fine-tuned on MSCOCO.**
> >
> > Non-LLM models like GIT [1] are typically task-specific and do not possess instruction-following capabilities. Consequently, they cannot respond to novel queries outside their training data, nor can they effectively leverage image-grounded guidance to mitigate hallucinations without special adaptations.
> >
> > We emphasize that MARINE, like VCD [2], is specifically designed for **the decoding stage of LVLMs**. Their effectiveness relies on these models' ability to incorporate guidance into the generation process. We appreciate your suggestion and believe that further research into applying MARINE to various models could be valuable. However, given the design and capabilities of LVLMs versus task-specific models fine-tuned on restricted datasets like MSCOCO, we believe that the scope of MARINE reasonably lies in improving models with broader instruction-following abilities.
> >
> > [1] GIT: A Generative Image-to-text Transformer for Vision and Language
> >
> > [2] VCD: Mitigating Object Hallucinations in Large Vision-Language Models through Visual Contrastive Decoding

---

> > > ### Comment · Reviewer_YMx7 · 2024-11-21
> > > **Thank the authors for your response**
> > >
> > > Thank you for your thorough responses. The clarifications and additional experiments have helped better contextualize the work's contributions and limitations.
> > >
> > > The response has effectively addressed several concerns:
> > > - Part 3 provides clear clarifications with specialized image captioning models
> > > - The solution to solve vocabulary gap is reasonable
> > >
> > > Meanwhile, results from our discussion in Parts 1 and 2 show some limitations remain:
> > >  1. The method's effectiveness appears inversely correlated with model capability - showing stronger gains on weaker models like LLaVA but diminishing returns on more advanced models
> > > 2.  For LLaVA, while improvements are shown, the performance gap is substantial compared to specialized image captioning models
> > > 3. With stronger models (e.g., mPLUG-Owl2), more sophisticated approaches like dynamic guidance weight did not yield significant improvements
> > >
> > > The method offers practical advantages (e.g no training required). However, its primary utility appears limited to improving baseline models rather than advancing the state-of-the-art. Given these considerations regarding technical novelty and scope of applicability, I maintain my original rating, acknowledging the paper's contribution while noting its limitations.
> > >
> > > Thank again for the constructive discussion with the authors.

---

> ### Author Response · Authors · 2024-11-22
> **Further clarifications on the remaining concerns (Part 1)**
>
> Thank you very much for your prompt feedback and for acknowledging that our rebuttal has addressed several of your concerns. We appreciate your time in this discussion, and would like to respectfully address some **significant misunderstandings** regarding the scope and contribution of our work. Below, we provide detailed clarifications:
>
> ---
>
> ### Scope and contribution of MARINE
>
> In response to the limitation you mentioned, we must clarify that
> - **Primary focus**: The focus of our method MARINE is to **reduce object hallucinations of LVLMs** for their diverse responses to a wide range of tasks, not to outperform specialized image captioning models on captioning. This focus is emphasized in lines 39-53 and further discussed in lines 112-135. Object hallucination reduction is a critical area of research in the LVLM domain, as demonstrated by numerous recent studies.
> - **Baselines (recent works)**: As outlined in our work, the appropriate comparison for MARINE is against the recent **methods targeting hallucination reduction in LVLMs** such as LURE [1], Woodpecker [2] and VCD [3]. Comparing MARINE to specialized image captioning models, which address fundamentally different tasks, misrepresents the scope and contribution of our work.
> - **State-of-the-art performance**: As shown in both **Table 1 and 2** of our manuscript, MARINE consistently achieves SOTA performance across multiple metrics when compared to other methods designed to mitigate hallucinations in LVLMs.
>
> ---
>
> ### Clarifications on image captioning experiments (Bleu, ROUGE-L, CIDEr)
>
> The evaluation of image captioning in our paper is *not the focus of MARINE* but serves a complementary purpose. These additional experiments demonstrate that MARINE can reduce hallucinations while maintaining overall task performance across diverse applications. This is consistent with related works [1-3], which also focus on hallucination reduction rather than specialized image captioning.
>
> We have similarly pointed this out in our previous rebuttal:
>
> > “As we previously pointed out in the paper (line 414-416), image captioning evaluation serves to demonstrate that our method, while primarily targeting hallucination mitigation, maintains LVLMs' performance on broader tasks without significant trade-offs.”
>
> > “The competitive performance of LVLMs, combined with their flexibility across multiple tasks, supports our focus on enhancing these models rather than specialized ones.”
>
> In addition, as demonstrated in Figure 4, 7, 9 and 13, our paper includes question examples of our targeted evaluation that are fundamentally different from image captioning tasks, such as:
>
> - (Figure 4) Is there a chair in the image?
> - (Figure 4) What is the position of the skateboard in the image?
> - (Figure 7) What might be the purpose of this table arrangement?
> - (Figure 9) How might the art installation encourage interaction among people?
> - (Figure 13) What precautions has the little girl taken to stay dry in the rain?
>
> While the question format of one metric CHAIR may resemble image captioning in format, they specifically assess **the degree of object hallucination in LVLMs** as compared the captioning metrics like CIDEr. The objective is not to have models generate captions that mimic ground truth but to reduce hallucinations in LVLMs **across diverse tasks and response types**.

---

> ### Author Response · Authors · 2024-11-22
> **Further clarifications on the remaining concerns (Part 2)**
>
> ### Addressing specific follow-up comments
>
> ---
>
> **Q1: The method's effectiveness appears inversely correlated with model capability.**
>
> We respectfully disagree. We respectfully disagree. For example, using the POPE metric, MARINE significantly improves the performance of mPLUG-Owl2:
> - Accuracy: **72.5\% → 82.8\%**
> - F1 score: **77.5\% → 82.7\%**
> - Yes ratio: Improved from **72.4\% → 49.2\%**, aligning closely with the ideal ratio of 50\%.
>
> ---
>
> **Q2. For LLaVA, while improvements are shown, the performance gap is substantial compared to specialized image captioning models.**
>
> As emphasized earlier, MARINE’s focus is **not on image captioning** but on reducing object hallucinations in LVLMs. Image captioning evaluations are provided to demonstrate MARINE's broad applicability without trade-offs, not to claim superiority over specialized models.
>
> ---
>
> **Q3. With stronger models (e.g., mPLUG-Owl2), more sophisticated approaches like dynamic guidance weight did not yield significant improvements.**
>
> Dynamic guidance weight is an **optional enhancement**, not the core of MARINE. The modest improvement compared to the vanilla MARINE approach highlights **the robustness and effectiveness of our original method**, which performs strongly even without additional complexity. For your clearer reference, we added the original performance of mPLUG-Owl2 below, with subscripts indicating the performance gain compared to greedy decoding:
>
> | Method                     | LLAVA-$C_s\downarrow$ | LLAVA-$C_i\downarrow$ | LLAVA-$Recall\uparrow$ | mPLUG-Owl2-$C_s\downarrow$ | mPLUG-Owl2-$C_i\downarrow$ | mPLUG-Owl2-$Recall\uparrow$ |
> |----------------------------|-----------------------|-----------------------|------------------------|---------------------------|---------------------------|-----------------------------|
> | Greedy                     | 26.6                 | 10.5                 | 47.4                  | 6.2                       | 3.4                       | 38.8                       |
> | Fix Guidance Strength      | 17.8$_{-8.8}$        | 7.2$_{-3.3}$         | 50.8$_{+3.4}$         | 4.2$_{-2.0}$              | 2.3$_{-1.1}$              | 41.4$_{+2.6}$              |
> | Dynamic Guidance Strength | 14.8$_{-11.8}$   | 6.5$_{-4.0}$     | 49.9$_{+2.5}$         | 5.0$_{-1.2}$              | 2.6$_{-0.8}$              | 41.0$_{+2.2}$              |
>
> | Method                     | LLAVA-$Accuracy\uparrow$ | LLAVA-$F1\uparrow$ | LLAVA-Yes Ratio | mPLUG-Owl2-$Accuracy\uparrow$ | mPLUG-Owl2-$F1\uparrow$ | mPLUG-Owl2-Yes Ratio |
> |----------------------------|--------------------------|--------------------|-----------------|-----------------------------|------------------------|-----------------------|
> | Greedy                     | 51.8                    | 67.4               | 97.7            | 72.5                        | 77.5                   | 72.4                  |
> | Fix Guidance Strength      | 66.9$_{+15.1}$          | 72.9$_{+5.5}$      | 72.3$_{-25.4}$  | 82.8$_{+10.3}$              | 82.7$_{+5.2}$          | 49.2$_{-23.2}$        |
> | Dynamic Guidance Strength | 71.97$_{+20.2}$     | 74.48$_{+7.1}$ | 59.83$_{-37.9}$ | 83.3$_{+10.8}$          | 83.2$_{+5.7}$      | 49.4$_{-23.0}$    |
>
> We hope these clarifications address your concerns. Please reach out if further explanations are needed. We appreciate your efforts in reviewing our work and look forward to continued discussion.
>
> ---
>
> [1] Analyzing and mitigating object hallucination in large vision-language models.
>
> [2] Woodpecker: Hallucination correction for multimodal large language models.
>
> [3] Mitigating Object Hallucinations in Large Vision-Language Models through Visual Contrastive Decoding.

---

> > ### Comment · Reviewer_YMx7 · 2024-11-26
> > **Clarification on review comments**
> >
> > I appreciate the further clarifications. Let me summarize the aligned understanding:
> > 1. The focus of the work is hallucination reduction. The SOTA is defined within this particular task, measured by CHAIR and POPE metrics.
> > 2. Under this scope, the proposed method improves the hallucination metrics when applied on different base models.
> > 3. Image captioning metrics and dynamic guidance weight is not the key of this paper.
> >
> > While these points are well taken and reflected in the current assessment, I would like to articulate the key considerations behind my review. The major concern lies in the technical contribution of this work:
> > 1. The method of using detection results to adjust the token logits is straightforward and not novel.
> > 2. The method has a relatively narrow focus on the object hallucination issue, while the overall quality of LVLMs is evaluated in multiple aspects (instruction understanding, visual understanding, reasoning, etc [1])
> > 3. State-of-the-art LVLMs (in a general sense, not defined by CHAIR and POPE metrics) like GPT-4V already demonstrate strong performance across aspects including object accuracy
> >
> > To be constructive, the proposed work can still contribute to the field meaningfully if:
> > 1. The proposed work improves LVLMs in object hallucination, while not compromising other metrics (e.g. image captioning)
> > 2. The method can be extended and inspire further research. Dynamic guidance strength was mentioned as an optional example.
> >
> > Based on recent discussions,MARINE indeed improves both hallucination and image captioning performance for LLaVA, which is good. However, such effectiveness is not seem on stronger base models like mPLUG-Owl2, where MARINE improves hallucination metrics with some trade-off in image captioning performance. Therefore, the method's scalability to SOTA base models (in a general sense, not particularly in object hallucination), which are more commonly used in AI applications, remains uncertain.
> >
> > The review recommendation is therefore based on balancing the method's effectiveness in its claimed scope against its technical contribution to the broader field.
> >
> > Thank the authors again for the constructive discussion.
> >
> > [1] Ye, Qinghao, et al. "mplug-owl: Modularization empowers large language models with multimodality." arXiv preprint arXiv:2304.14178 (2023).

---

> > > ### Author Response · Authors · 2024-11-26
> > > **Thank you for the continued discussion**
> > >
> > > Thank you for your continued engagement and thoughtful feedback. We are pleased to have reached consensus on the SOTA performance of our method in mitigating object hallucination, the core focus of our work. Below, we hope to further discuss your remaining concerns:
> > >
> > > > The method has a relatively narrow focus on the object hallucination
> > >
> > > Mitigating hallucination is a critical research area for LLMs and LVLMs, directly tied to their **reliability and trustworthiness**. Hallucination reduction aligns with the "*honesty*" aspect of LLM training objectives, as highlighted by OpenAI and Anthropic [1, 2], which complements other key goals like *helpfulness* (reasoning) and *harmlessness* (safety). While the aspect you mentioned is indeed important and relates to the helpfulness of LLMs, the two other aspects remain equally significant for ensuring the trustworthy and benign development of AI.
> > >
> > > For LVLMs specifically, object hallucination is a distinct issue arising from the integration of visual modules. The significance of this subfield is well-supported by works like [3-7], which **also focused on metrics such as CHAIR and POPE** to address this challenge. For example,
> > > - Table 1 of [3], Table 1 of [6], Table 1&2 of [7] considers CHAIR
> > > - Table 1 of [4], Table 1 of [5], Table 2 of [6]  considers POPE
> > >
> > > Along the same direction of this body of work, our approach targets a well-recognized and important problem.
> > >
> > > > State-of-the-art LVLMs like GPT-4V already demonstrate strong performance across aspects including object accuracy
> > >
> > > We acknowledge that GPT-4V demonstrates strong object accuracy across multiple tasks. However, the broader research community is focused on advancing **open-source** LVLMs, which remain susceptible to hallucinations. Closed-source models like GPT-4V, while having superior performance, are inaccessible for widespread research and applications requiring transparency.
> > >
> > > Additionally, we recognize the computational resource limitations within academia, where running experiments with models larger than 14B parameters is often challenging and requires significant GPU resources that may not be available. While the best-performing models are typically large (>70B and GPT-4 >200B), much of the research in this field still focuses on smaller models (7B-14B), which remain a meaningful and significant area of study.
> > >
> > > Our work contributes by improving open-source models such as LLaVA and mPLUG-Owl. This is analogous to active efforts in other areas, such as improving mathematical reasoning in open-source LLMs like Llama, even though GPT-4o currently exhibits near-upper-bound performance in this area. These contributions advance the state of open research, propose effective algorithms, and inspire further innovations.
> > >
> > > > The method of using detection results to adjust the token logits is straightforward and not novel.
> > >
> > > Regarding the perceived simplicity of our approach, our method represents one of the first to leverage visual guidance for mitigating object hallucinations. Concurrently, similar frameworks for controllable generation are emerging [5-6]. Lines 127-135 of our paper discuss the differences of these works with ours in detail.
> > >
> > > Furthermore, we agree that extending our approach to ensure no trade-offs in other metrics, such as image captioning, is crucial for future research and broader impact. While our results for LLaVA demonstrate such dual improvements, extending to achieve consistent outcomes across all models remains an important direction for future work.
> > >
> > > We sincerely thank you for your constructive suggestions and the opportunity to address these points. We hope this additional context clarifies our contributions and the broader significance of our work. Please let us know if you have any further questions.
> > >
> > > ---
> > >
> > > [1] Training language models to follow instructions with human feedback.
> > >
> > > [2] A General Language Assistant as a Laboratory for Alignment
> > >
> > > [3] Analyzing and mitigating object hallucination in large vision-language models.
> > >
> > > [4] Woodpecker: Hallucination correction for multimodal large language models.
> > >
> > > [5] Mitigating Object Hallucinations in Large Vision-Language Models through Visual Contrastive Decoding.
> > >
> > > [6] Paying more attention to image: A training-free method for alleviating hallucination in lvlms
> > >
> > > [7] OPERA: Alleviating Hallucination in Multi-Modal Large Language Models via Over-Trust Penalty and Retrospection-Allocation

---

> > > > ### Author Response · Authors · 2024-11-29
> > > >
> > > > Dear reviewer YMx7,
> > > >
> > > > We are writing to kindly follow up on our previous response to your feedback. We have addressed the key concerns raised and provided additional context to clarify the significance and contributions of our work.
> > > >
> > > > If there are any remaining points you would like us to address, we would be happy to provide further clarification. We look forward to hearing your thoughts.

---

### Official Review · Reviewer_7QRb · 2024-11-03

**Soundness:** 3
**Presentation:** 2
**Contribution:** 2
**Rating:** 6
**Confidence:** 4

**Summary:**

The paper introduces MARINE, a novel framework to reduce hallucinations in Large Vision-Language Models (LVLMs) without requiring additional training or API access. MARINE provides image-grounded guidance by leveraging open-source vision models (like object detectors) to extract object-level information. This approach enhances the accuracy of LVLM-generated content by integrating multiple vision models for reliable object-level guidance. Extensive evaluations on five LVLMs show that MARINE effectively reduces hallucinations and outperforms fine-tuning methods, maintaining the detail and precision of LVLM outputs.

**Strengths:**

1. MARINE novelly utilizes the strengths of image-grounded visual models (like object detectors) to provide detailed information about the input image, effectively helping to reduce hallucinations in LVLMs.

2. Extensive evaluations across multiple datasets show that MARINE consistently outperforms baseline methods in mitigating hallucinations, while preserving strong performance across various tasks, including image captioning and visual question answering.

**Weaknesses:**

1. Would it be possible to revise the input text prompt to "enforce" LVLMs to only describe the significant objects to mitigate hallucination? Would it achieve similar results?  It is recommended to compare to a baseline using carefully engineered prompts aimed at reducing hallucination. This would allow for a more concrete evaluation of MARINE versus prompt engineering approaches.

An example of a prompt is "Describe the visible contents of this image in as much detail as possible without adding any information not clearly visible. Only mention objects, colors, shapes, and textures that can be directly observed in the image, avoiding assumptions about materials, functions, or contexts. If there are any uncertainties about what an object is, describe its visual characteristics (e.g., 'a circular object with a smooth surface') without inferring its purpose or identity. Avoid creative or hypothetical descriptions, and focus on observable details only." Other prompts for a Vision-Language Model (VLM) that minimizes hallucination, focuses on clear, fact-based, and constraint-driven descriptions, are also reasonable.

2. Since the hallucination mitigation is achieved by resembling scores coming from the Guidance Model and LVLM, would it be similar to only conducting an ensemble using different LVLM? I wonder what is the significance and novelty of introducing external image-grounded visual models guidance models. Also, would it achieve similar results by increasing the threshold of object scores so that models won't hallucinate objects with low uncertainty? They can serve as additional baselines to compare with.

**Questions:**

1. I wonder how much this hallucination problem can be attributed to the visual encoder used by LVLMs. I think MARINE is using other visual models to provide more semantics, or information, in the input prompt, to enhance the downstream tasks to reduce mitigation. Would the results be similar by simply replacing the visual encoder of LVLMs with those image-grounded visual models (like DINO, SAM, or DETR)? The authors could conduct an ablation study comparing MARINE to LVLMs with different visual encoders, including the image-grounded models used in MARINE. This would help isolate the impact of the visual encoder versus the guidance approach.

---

> ### Author Response · Authors · 2024-11-21
> **Rebuttal to Reviewer 7QRb - Part 1**
>
> Thank you very much for your support and the constructive feedback that helped us improve our work. Please see our detailed response with additional experiments below.
>
> ---
>
> ### W1. Compare to a baseline using carefully engineered prompts.
> Thank you for your insightful suggestion to compare our approach against a baseline using carefully engineered prompts aimed at reducing hallucination. Based on your recommendation, we conducted additional experiments and present the results in the table below and Table 18 of our revision:
> | Method  | LLAVA-$C_s\downarrow$ | LLAVA-$C_i\downarrow$ | LLAVA-$Recall\uparrow$ | LLaVA-v1.5-$C_s\downarrow$ | LLaVA-v1.5-$C_i\downarrow$ | LLaVA-v1.5-$Recall\uparrow$ | mPLUG-Owl2-$C_s\downarrow$ | mPLUG-Owl2-$C_i\downarrow$ | mPLUG-Owl2-$Recall\uparrow$ |
> |---------|---------|---------|----------|--------|--------|----------|---------|--------|----------|
> | Original  | 26.6        | 10.5        | 47.4         | 8.8              | 4.6              | 41.1             | 6.2                       | 3.4                       | 38.8                      |
> | Direct Prompting          | 27.2        | 11.0        | 46.4         | 19.6             | 8.3              | **52.3**             | 9.0              | 5.1              | **42.0**             |
> | Prompts as Additional Guidance | 37.4        | 10.5        | 50.4         | 12.6             | 5.9              | 44.6             | 6.6              | 3.9              | 40.4             |
> | **MARINE** (ours)  | **17.8**    | **7.2**     | **50.8**     | **6.2**          | **3.0**          | 44.3         | **4.2**          | **2.3**          | 41.4         |
>
> Specifically, we evaluated:
> - **Direct Prompting**: The original input query was replaced with the prompts as described.
> - **Prompts as Additional Guidance**: We incorporated the prompt as supplemental context to guide the models in generating outputs.
> The results demonstrate that incorporating the prompt can effectively enhance the recall performance for some models (e.g., LLaVA-v1.5 improves from 41.1 to 52.3 in recall). However, it did not consistently reduce hallucinations across all metrics and the performance on CHAIR scores (e.g., $C_s$, $C_i$) dropped. Meanwhile, MARINE significantly outperforms the prompting baseline approaches on CHAIR.
>
> We note the following differences between our method and prompting method:
> - The Prompting method depends heavily on the instruction-following ability of the model. While it might mitigate the hallucination to a mild extent for strong models (e.g., LLaVA-v1.5), it may cause a weak model to hallucinate even more (e.g., LLaVA). Models also require more sophisticated fine-tuning approaches to generate better and more precise responses conditioned on the prompts, as discussed in [1]. In contrast, our method directly addresses deficiencies in the model’s vision capabilities by introducing stronger vision guidance. This makes our approach more effective even for weaker models and more cost-efficient.
> - Unlike prompting methods, which need to be tailored to specific tasks or datasets, our method generalizes effectively across models and datasets, reducing hallucinations while maintaining competitive recall.
>
> [1] Enhancing Large Vision Language Models with Self-Training on Image Comprehension

---

> > ### Author Response · Authors · 2024-11-21
> > **Rebuttal to Reviewer 7QRb - Part 2**
> >
> > ### W2. Would it be similar to only conducting an ensemble using different LVLM?
> > Thank you for your valuable insight. Ensembling different LVLMs is indeed a strong baseline, as majority voting among models enhances robustness. However, it is less frequently used in practice due to the significant cost of acquiring and loading multiple LVLMs, which is substantially higher than that of additional vision models.
> >
> > In response to your question, we conducted experiments by **ensembling all possible combinations of the five LVLMs** considered in our paper on the POPE benchmark. The average performance is summarized below and updated in Appendix D.1.2 with Table 19:
> > | Method   | $Accuracy\uparrow$           | $F1\uparrow$            | Yes Ratio           |
> > |----------|----------|-----------|---------------|
> > | Voting on 2 LVLMs | 76.7 ± 7.8    | 79.8 ± 4.8    | 62.6 ± 11.2   |
> > | **MARINE (ours)**  | **79.9** ± 7.4    | **80.4** ± 4.6    | **51.1** ± 12.6   |
> > |-||
> > | Voting on 3 LVLMs       | 83.2 ± 1.2    | 83.0 ± 1.0    | 48.6 ± 2.8    |
> >
> > Notably, our method outperforms the ensemble of two LVLMs. While the ensemble of three LVLMs achieves higher scores, it comes with significantly higher computational and memory costs, making it much less practical for many real-world scenarios. As further shown below, our method requires only 30% more GPU memory than the plain LVLM during inference:
> >
> >  | Metric        | Greedy  | MARINE (Ours) |
> > |-----------|-------------|--------------------|
> > | Peak GPU Memory Usage during Inference (GB) | 23.53       | 30.78 (x1.30)      |
> >
> > For large-batch inference and online chatbot deployment, ensembling multiple LVLMs is much less feasible due to the substantial increase in memory consumption. In contrast, MARINE remains an accessible, efficient, and effective approach.
> >
> > Furthermore, since most LVLMs use the same CLIP model as their vision component, ensembling multiple LVLMs primarily combines the language outputs, aiming for consistency across **different LLMs**, which is a strategy proven effective in the textual domain. This makes the ensemble of different LVLMs **complementary to our method**, which focuses on enhancing the vision component by incorporating multiple vision models.
> >
> > Lastly, while ensembling LVLM outputs for yes/no questions in POPE is straightforward, extending this approach to a diverse range of tasks—including instruction-following tasks—is significantly more challenging. Developing a feasible and effective method for ensembling LVLMs in such contexts remains an unexplored area for future research. In contrast, MARINE easily generalizes to different question formats and tasks, demonstrating its versatility and practicality.
> >
> > ---
> >
> > ### W3. Add additional baselines by increasing the threshold of object scores to mitigate object hallucinations
> >
> > While our approach enables adjusting the threshold for object scores, implementing this in vanilla LVLMs is challenging because they lack explicit mechanisms to determine whether the next token corresponds to an object, making threshold adjustments difficult.
> > Similar strategies have been explored in related works such as LURE [1] and OPERA [2]. LURE addresses hallucinations by masking low-confidence words and fine-tuning MiniGPT-4, effectively filtering out uncertain object descriptions. OPERA reduces overconfidence during beam search by penalizing logits and adjusting token selection to mitigate hallucinations without additional training.
> >
> > We have included these methods as baselines in our evaluation (see Tables 1 and 2), and our approach significantly outperforms them across various metrics. This suggests that while adjusting thresholds can provide some benefits, our method's ability to aggregate guidance from multiple image-grounded models offers a more robust and effective solution for mitigating hallucinations by leveraging consensus among models to enhance factual accuracy.
> >
> > [1] Analyzing and Mitigating Object Hallucination in Large Vision-Language Models
> >
> > [2] OPERA: Alleviating Hallucination in Multi-Modal Large Language Models via Over-Trust Penalty and Retrospection-Allocation

---

> > > ### Author Response · Authors · 2024-11-21
> > > **Rebuttal to Reviewer 7QRb - Part 3**
> > >
> > > ### Q1. Would the results be similar by replacing the visual encoder of LVLMs with those image-grounded visual models (like DINO, SAM, or DETR)?
> > >
> > > Thank you for raising this question. Current LVLMs predominantly use CLIP as their vision backbone because it is trained on large image-text corpus, achieving strong alignment between visual and textual modalities, which is a crucial foundation for training LVLMs. In contrast, object grounding models like DETR specialize in object detection tasks and lack this textual alignment. Using them as the sole vision backbone may harm the LVLM's instruction-following ability due to their limited capacity to connect visual features with language.
> > >
> > > Training an LVLM with a different vision backbone from scratch is further resource-intensive. It requires training an alignment layer to connect the new vision backbone to the LLM and extensive post-training to enable the model's instruction-following capabilities. For example, training LLaVA-v1.5 involved 665K vision-language instruction-following data and required an 8-A100 GPU node.
> > >
> > > Due to computational constraints, we experimented with directly and solely using visual guidance from DETR and removed the soft prompt from CLIP. In image captioning tasks, the model either struggled to process the image or produced hallucinations. For example:
> > > - *“Sure, I'd be happy to help! Can you please provide the image or a link to the image you would like me to caption?”*
> > > - *“Sure, I can help you with that! Based on the image you provided, which features a teddy bear and a bed, I would suggest the following caption:\n\n\"A cozy bed with a cuddly teddy bear, creating a warm and inviting atmosphere for a peaceful night”*
> > >
> > > These results indicate that direct replacement requires additional tuning to align new visual encoders with the LLM, and having the original visual encoder trained by CLIP is necessary for the current LVLM implementations.

---

> > > > ### Author Response · Authors · 2024-11-23
> > > > **Invitation for discussion**
> > > >
> > > > Dear reviewer 7QRb,
> > > >
> > > > Thank you again for your support and thoughtful feedback. We hope our responses and additional experiments have addressed your questions comprehensively. Specifically:
> > > >
> > > > - W1: We evaluated direct prompting baselines and demonstrated MARINE’s superior performance in reducing hallucinations and enhancing generalization while training-free.
> > > > - W2: We added an ensemble baseline of multiple LVLMs, highlighting its strengths and practical limitations compared to MARINE. We further discussed thresholding object scores and compared our method to related strategies, showcasing MARINE’s robustness.
> > > > - Q1: We clarified the challenges of replacing LVLM visual encoders and provided experimental insights.
> > > >
> > > > Please let us know if you have further questions or require additional details. We deeply appreciate your time and effort in reviewing our work.

---

### Author Response · Authors · 2024-11-21
**Global comment**

We sincerely thank all the reviewers for their insightful and encouraging feedback on our manuscript. We appreciate the recognition of the effectiveness of our method (Reviewers 7QRb, YMx7, JbpG), and the simplicity and flexibility of our approach (Reviewers JbpG, 8B3T, YMx7). We also thank the reviewers for noting the clarity of our method description (Reviewers HeC9, JbpG) and the comprehensiveness of our experiments demonstrating consistent performance improvements (Reviewers 7QRb, YMx7, 8B3T). Finally, we appreciate the recognition of our efforts in enhancing the credibility of our work by providing code and all evaluation outputs (Reviewer 8B3T).

In response to the comments, we provided additional experiments and revisions in the revised manuscript. Specifically, the extended results (page 21-25) include

- **Multiple additional baselines (Appendix D.1)**: Additional baselines include direct prompting (Table 18), ensembling different LVLMs (Table 19), and specialized captioning models GIT for image captioning task as a reference (Table 20). We continue to show the effectiveness and competitiveness of MARINE.
- **Dynamically adjusting the guidance strength (Appendix D.2, Table 21 and 22)**: We can dynamically adjust the guidance strength by the confidence score, which further provides marginal improvement..
- **Comprehensive latency and memory analysis (Appendix D.3, Table 23 and 24)**: Our method does not introduce significant computation overhead even if in the online setting of generating guidance during inference. The memory consumption only increase by $30\%$ instead of doubling.
- **Additional ablation study on guidance strength (Appendix D.4, Figure 8, 9 and Table 25)**: An overly strong guidance strength results in decayed performance of LVLMs in instruction-following tasks.

We have also addressed the comments in detail in each individual rebuttal.

---

### Meta-Review · Area_Chair_oUB9 · 2024-12-19

**Metareview:**

This paper proposed a training-free and API-free method named MARINE to address object hallucinations in LVLMs. MARINE efficiently reduces object hallucinations by making use of image-grounded guidance. Specifically, it applies publicly available object detection models to get visible object information from the image and uses it as control guidance to adjust token logits in the output of LVLM. The proposed method was evaluated on several LVLMs with multiple tasks demonstrating its effectiveness in reducing object hallucinations.

The strengths of this paper are as follows: 1. As acknowledged by reviewers, applying image-grounded models to reduce object hallucination in LVLM is a novel and promising direction; 2. Extensive experiments demonstrate the effectiveness of the proposed methods. However, as pointed out by reviewer YMx7, the major concerns about this paper is that the technical contribution is limited. Applying detection results to adjust output logits of LVLMs is straightforward and not novel, which provides limited insights to the broader field.

**Additional Comments On Reviewer Discussion:**

Reviewer 7QRb, 8B3T, JbpG and YMx7 all acknowledge that the proposed method is simple and effective. During discussion, the authors fully addressed concerns proposed by 7QRb, 8B3T. Reviewer JbpG still has some concerns, however, reviewer JbpG did not mention which concerns need to be further addressed.

Reviewer YMx7 has a lot of discussion with the authors. The authors try to address the concerns raised by YMx7 carefully by including more experiments and explanations. However, this does not fully address the concerns raised by reviewer YMx7. Reviewer YMx7 admits that the proposed method is effective, however, the technical contribution is limited which I agree. As in the discussion, current closed-source LVLMs such as GPT-4V already achieved great results (which admitted by the author) in reducing object hallucination, the proposed method cannot achieve state-of-the-art results. I agree with the arguments from the authors that they focus more on reducing object hallucinations from open-sourced models and it is a bit unfair to compare with closed-sourced models. However, due to the existence these closed-sourced models, the technical contributions are more important as they can provide more insight. Reviewer YMx7  also provide constructive suggestions to further improve this paper so that this paper could inspire future research.

Reviewer HeC9 gives highest score for this paper. However, HeC9 misunderstands the paper. This paper does not apply 2D visual feature  to LVLM, instead, it applies the output from object detection model as the conditional guidance.

Considering all the discussions, I agree this paper proposed a simple and effective method to address object hallucination in LVLM, however, technical contributions are limited and the proposed methods need to be improved to benefit the future research.

---

### Decision · Program_Chairs · 2025-01-22

Reject